# MACHINE TEXT DETECTORS ARE MEMBERSHIP INFERENCE ATTACKS

## ABSTRACT

Although membership inference attacks (MIAs) and machine-generated text detection target different goals, identifying training samples and synthetic texts, their methods often exploit similar signals based on a language model's probability distribution. Despite this shared methodological foundation, the two tasks have been independently studied, which may lead to conclusions that overlook stronger methods and valuable insights developed in the other task. In this work, we theoretically and empirically investigate the *transferability*, i.e., how well a method originally developed for one task performs on the other, between MIAs and machine text detection. For our theoretical contribution, we prove that the metric that achieves the asymptotically highest performance on both tasks is the same. We unify a large proportion of the existing literature in the context of this optimal metric and hypothesize that the accuracy with which a given method approximates this metric is directly correlated with its transferability. Our large-scale empirical experiments, including 7 state-of-the-art MIA methods and 5 state-of-the-art machine text detectors across 13 domains and 10 generators, demonstrate very strong rank correlation ($\rho > 0.6$) in cross-task performance. We notably find that Binoculars, originally designed for machine text detection, achieves state-of-the-art performance on MIA benchmarks as well, demonstrating the practical impact of the transferability. Our findings highlight the need for greater cross-task awareness and collaboration between the two research communities. To facilitate cross-task developments and fair evaluations, we introduce MINT, a unified evaluation suite for MIAs and machine-generated text detection, with implementation of 15 recent methods from both tasks.[1]

## 1 INTRODUCTION

Large language models (LLMs) have demonstrated human-level generative and understanding capabilities, impacting fields such as creative writing (Chakrabarty et al., 2024), news reporting (Futurism, 2023), and even scientific discovery (Lu et al., 2024). Despite many positive societal implications, their negative consequences have increasingly been reported. For instance, LLMs may leak personal (Lukas et al., 2023) or copyrighted information (Wei et al., 2024) due to their memorization of training data (Morris et al., 2025; Carlini et al., 2021). In addition to privacy issues, LLMs also raise challenges to authorship authenticity, as they can be exploited for mass-producing propaganda (Goldstein et al., 2024) or cheating on student assignments (Guardian, 2025).

Many recent works aim to mitigate such negative implications of LLMs. One major direction is membership inference attacks (MIAs), which attempt to classify whether a given text sample is a member of the training data of a language model (Carlini et al., 2022; Mattern et al., 2023; Shi et al., 2024). This helps identify potential leaks of personal information and copyright infringement. To address the challenges of authorship authenticity, another major line of work is machine-generated text detection, which distinguishes between human-written and machine-generated texts (Ippolito et al., 2020; Mitchell et al., 2023; Hans et al., 2024; Yang et al., 2024). This safeguards against misuse of LLMs (e.g., misinformation and academic misconduct) by flagging suspicious texts.

While these two tasks, membership inference and machine text detection, target different goals, their methods often leverage similar signals based on a language model's probability distribution.

---

[1] https://anonymous.4open.science/r/ICLR_repo-520C

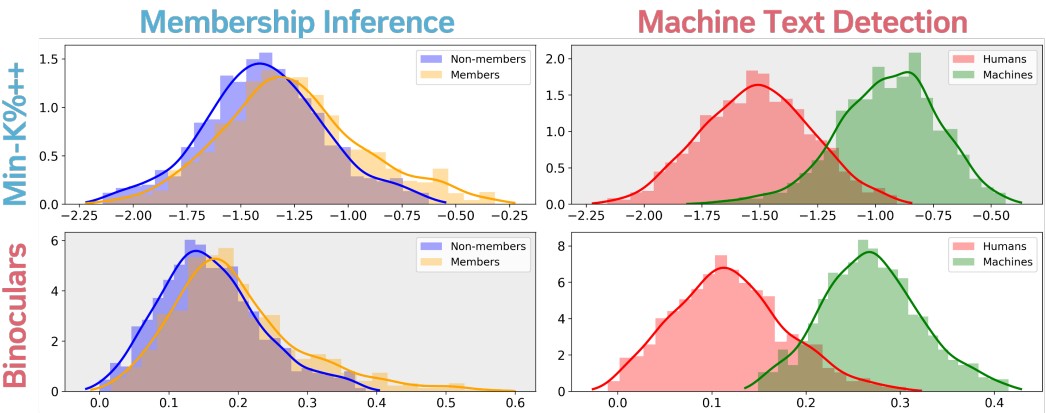

Figure 1: **Predicted score distributions** of Min-K%++ (state-of-the-art MIA) and Binoculars (state-of-the-art machine text detector) on both tasks. Shaded areas indicate the cross-task setting. We see that, although Binoculars and Min-K%++ were developed for two separate tasks, the distributions across populations induced by both metrics are **strikingly similar**, suggesting their transferability.

In membership inference, documents that are members of a model's training dataset tend to show higher likelihood under the model than non-members. Similarly, in machine text detection, machine-generated texts typically exhibit higher likelihoods under the target model than human-written texts. Thus, both tasks use text likelihood or entropy as standard baselines. Notably, Neighborhood attack (Mattern et al., 2023) from membership inference and DetectGPT (Mitchell et al., 2023) from machine text detection have been noted as essentially identical (Shi et al., 2024; Naseh & Mireshghallah, 2025). Both estimate the probability curvature around a target text by perturbing the text. Moreover, Figure 1 shows strikingly similar score distributions of the state-of-the-art MIA and detector on both tasks. Despite this broader shared methodological foundation, the two tasks have been independently studied. This may result in biased evaluations that overlook stronger methods developed for one task and consequently lead to conclusions that miss valuable insights from the other.

Motivated by this gap, we theoretically and empirically study the *transferability*, i.e., how well a method originally developed for one task performs on the other, between MIAs and machine text detection. For our theoretical contribution, we prove that both tasks share the same optimal metric for achieving asymptotically highest performance: *the likelihood ratio test between the target model distribution and the true population distribution* (§2.3). We unify a large proportion of the existing methods from both tasks as approximations of this optimal metric and hypothesize that the transferability of a particular method is correlated with how well it approximates this metric.

To empirically quantify the transferability, we conduct large-scale experiments, spanning 7 state-of-the-art MIA methods and 5 state-of-the-art machine text detectors across 13 domains and 10 generators (§3.1). For each method, we evaluate its performance on both tasks, MIAs and machine text detection, and then compute the rank correlation between their performance rankings on the two tasks. We find that many methods effective in MIAs remain effective in machine text detection, and vice versa, with a strong rank correlation ($\rho > 0.6$) in their cross-task performance (§3.2). Furthermore, we notably find that Binoculars (Hans et al., 2024), originally designed for machine text detection, achieves state-of-the-art performance on MIAs as well, demonstrating the practical impact of the transferability.

## 2 THEORETICAL TRANSFER BETWEEN MIA AND MGTD

### 2.1 TASK FORMULATION

Let $\mathcal{X}$ be the set of all token sequences and let $\mathcal{Q}$ be an oracle that models the "true" probability distribution $P_{\mathcal{Q}}$ of human-written text over $\mathcal{X}$. Let $\mathcal{M}$ be a language model inducing distribution $P_{\mathcal{M}}$ over $\mathcal{X}$ and $\mathcal{D}_{train} \subset \mathcal{X}$ denote the unknown training set used to fit $\mathcal{M}$. For a given text $x \in \mathcal{X}$

our two tasks have the following hypotheses:

$$\textbf{Machine-Generated Text Detection} - H_0 : x \sim P_{\mathcal{Q}}, \;\; H_1 : x \sim P_{\mathcal{M}}$$

$$\textbf{Membership Inference} - H_0 : x \in \mathcal{X}, \;\; H_1 : x \in \mathcal{D}_{train}$$

Both tasks aim to develop some function $f(x; \mathcal{M})$ that can accept or reject their respective null hypotheses with maximum statistical power.

## 2.2 MOTIVATION: IDENTICAL PAIRS

We will start our discussion of theoretical transferability with a motivating example. Let $x = \{x_0, \cdots, x_n\}$ be a sequence of tokens and let $\mathcal{L}(x, \mathcal{M}) = -\sum_{x_i \in x} \log P_{\mathcal{M}}(x_i)$ be the surprisal of the sequence according to some model $\mathcal{M}$. The Neighborhood Attack (Mattern et al., 2023) is a membership inference method that is defined as follows

$$\textbf{Neighborhood}(x; \mathcal{M}; \phi) = \mathcal{L}(x; \mathcal{M}) - \frac{1}{n} \sum_{i=1}^{n} \mathcal{L}(\tilde{x}^{(i)}; \mathcal{M}) \tag{1}$$

Where $\{\tilde{x}^{(1)}, \cdots, \tilde{x}^{(n)}\} \sim \phi(x)$ is a set of perturbations $\tilde{x}^{(i)}$ that differ slightly from $x$ but are approximately equally likely to appear in the general distribution $\mathcal{X}$. In parallel to this, DetectGPT (Mitchell et al., 2023) is a popular and widely used machine-generated text detection method that is defined as follows

$$\textbf{DetectGPT}(x; \mathcal{M}; \phi) = \mathcal{L}(x; \mathcal{M}) - \mathbb{E}_{\tilde{x} \sim \phi(x)} \mathcal{L}(\tilde{x}; \mathcal{M})$$

We echo Naseh & Mireshghallah (2025) in noting that these two methods are approximately identical (the first is simply a finite sample approximation of the other). Given that this metric was proposed independently for both tasks it naturally invites further exploration into what other metrics may share this property. In the next section we provide a theoretical framework for understanding why this metric and others like it exhibit such strong performance on both tasks.

## 2.3 UNIFYING DETECTION AND MEMBERSHIP INFERENCE AS LIKELIHOOD RATIO TESTS

**Theorem 2.3 (Unified Optimality).** Let $\mathcal{X}$ be the set of all token sequences and let $\mathcal{Q}$ be an oracle that models the "true" distribution $P_{\mathcal{Q}}$ of human-written text over $\mathcal{X}$. Let $\mathcal{M}$ be a language model trained with cross-entropy to minimize the likelihood of some training set $\mathcal{D}_{train} \subset \mathcal{X}$. Then the test statistic

$$\Lambda(x) = \frac{\sum_{i=1}^{n} \log P_{\mathcal{M}}(x_i | x_{<i})}{\sum_{i=1}^{n} \log P_{\mathcal{Q}}(x_i | x_{<i})} = \frac{\mathcal{L}(x; \mathcal{M})}{\mathcal{L}(x; \mathcal{Q})}$$

achieves optimal accuracy at a given false positive rate (type I error) *for both machine-generated text detection and membership inference* under standard asymptotic regularity conditions, with coressponding maximum advantage (improvement over random guessing) bounded by

$$\text{adv} \leq \sqrt{\frac{D_{\text{KL}}(P_{\mathcal{Q}} \| P_{\mathcal{M}})}{8}}.$$

**Proof.** We prove this by showing that $\Lambda(x)$ coincides with the likelihood ratio test for both tasks.

*Step 1: Machine-generated text detection.* Consider testing

$$H_0 : x \sim P_{\mathcal{Q}}, \;\; H_1 : x \sim P_{\mathcal{M}}$$

The likelihood ratio test for this hypothesis is

$$\Lambda_{\text{MGT}}(x) = \frac{\mathcal{L}(x; \mathcal{M})}{\mathcal{L}(x; \mathcal{Q})}$$

which exactly coincides with the proposed statistic $\Lambda(x)$. By the Neyman-Pearson lemma (Neyman & Pearson, 1933), this test is uniformly most powerful for any given Type I error.

*Step 2: Membership inference.* Consider testing

$$H_0 : x \in \mathcal{X} \quad \text{versus} \quad H_1 : x \in \mathcal{D}_{\text{train}}.$$

Since $\mathcal{M}$ is trained via cross-entropy, it is a maximum likelihood estimator. Under standard asymptotic assumptions (sufficient model capacity, infinite training samples), $\mathcal{L}(x; \mathcal{M})$ converges to the likelihood of $x$ under $H_1$. Therefore, the likelihood ratio test for this hypothesis is also

$$\Lambda_{\text{MI}}(x) = \frac{\mathcal{L}(x; \mathcal{M})}{\mathcal{L}(x; \mathcal{Q})}$$

so $\Lambda(x)$ is also asymptotically optimal for membership inference.

*Step 3: Advantage bound.* For a binary hypothesis test with statistic $\Lambda(x)$, the Bayes error is

$$\varepsilon^* = \frac{1 - \text{TV}(P_{\mathcal{M}}, P_{\mathcal{Q}})}{2},$$

where $\text{TV}(\cdot, \cdot)$ denotes total variation distance. Applying Pinsker's inequality,

$$\text{TV}(P_{\mathcal{M}}, P_{\mathcal{Q}}) \leq \sqrt{\frac{1}{2} D_{\text{KL}}(P_{\mathcal{Q}} \| P_{\mathcal{M}})},$$

yields the stated bound on the error and the corresponding maximum advantage.

**Remark.** When $\mathcal{M}$ has sufficient capacity and is trained on asymptotically many samples from $\mathcal{D}_{train}$, the performance of the provided statistic $\Lambda(x)$ on membership inference will approach the optimal bound. However, since language models typically only get one pass through their training data in practice, it may be the case that other, more creative approximations of $P(x \in \mathcal{D}_{train})$ perform better than $P_{\mathcal{M}}(x)$ in real-world scenarios. Our result is not meant to discourage further work on these approximations, but rather to give an underlying theoretical framework for why the tasks of machine text detection and membership inference are so fundamentally connected.

**Discussion.** A key implication of this result is that any method that effectively approximates $\Lambda(x)$ will have high transferability, i.e., perform well on both machine-generated text detection and membership inference. In the remainder of this paper, we analyze a collection of state-of-the-art metrics using this framework and measure their degree of transferability in the context of this result.

### 2.4 Classifying Metrics as Approximate Likelihood Ratio Tests

Since the true population distribution of human-written text $P_{\mathcal{Q}}$ is inaccessible, methods in both tasks have employed various strategies to approximate this distribution. These strategies can be broadly divided into two approaches:

**Approximation via External Reference.** The first approach approximates the true distribution by leveraging an external distribution $P_{\mathcal{M}_{ref}}$ as a surrogate for the true population distribution $P_{\mathcal{Q}}$,

$$\mathcal{L}(x; \mathcal{Q}) \approx \mathcal{L}(x; \mathcal{M}_{ref}).$$

The surrogate distribution $P_{\mathcal{M}_{ref}}$ can be approximated via another language model (*Reference* (Carlini et al., 2021) from MIAs), a byte-level frequency distribution induced by Huffman encoding (*Zlib* (Carlini et al., 2021) from MIAs), a token-level frequency distribution from an external corpus (*DC-PDD* (Zhang et al., 2024) from MIAs), or cross-model entropy (*Binoculars* (Hans et al., 2024) from machine text detection).

**Approximation via Text Sampling.** The second approach leverages text sampling to approximate the true distribution $P_{\mathcal{Q}}$. The likelihood under the true distribution is approximated by the expected likelihood of multiple perturbations $\tilde{x}$ of the target text $x$,

$$\mathcal{L}(x; \mathcal{Q}) \approx \mathbb{E}_{\tilde{x} \sim \phi(\cdot|x)}[\mathcal{L}(\tilde{x}; \mathcal{M})],$$

where $\phi(\cdot|x)$ is a perturbation function that samples variations of the target text $x$. This strategy is employed by the *Neighborhood attack* (Mattern et al., 2023) from MIAs, *DetectGPT* (Mitchell et al., 2023), and *Fast-DetectGPT* (Bao et al., 2024) from machine text detection.

In Table 1 we provide a breakdown of the metrics we test in our work and where applicable include a reformulation to fit the categorization we propose. We discuss in more detail in Appendix A the steps we took to get to this general formulation for each example in the author's own notation.

| Metric | Task | Equational form |
|---|---|---|
| Reference (Carlini et al., 2021) | MIA | $\mathcal{L}(x;\mathcal{M}) - \mathcal{L}(x;\mathcal{M}_{ref})$ |
| Zlib (Carlini et al., 2021) | MIA | $\mathcal{L}(x;\mathcal{M}) \,/\, \text{Zlib}(x)$ |
| DetectLLM (Su et al., 2023) | Detection | $\mathbb{E}_{\tilde{x}\sim\phi(x)}[\mathcal{R}(\tilde{x};\mathcal{M})] \,/\, \mathcal{R}(x;\mathcal{M})$ |
| ReCall (Xie et al., 2024) | MIA | $\mathbb{E}_{\tilde{x}\sim\phi(x)}[\mathcal{L}(\tilde{x};\mathcal{M})] \,/\, \mathcal{L}(x;\mathcal{M})$ |
| DC-PDD (Zhang et al., 2024) | MIA | $\mathbb{E}_{\tilde{x}\sim\mathcal{M}}[\mathcal{L}(\tilde{x};\mathcal{M}_{ref})]$ |
| Binoculars (Hans et al., 2024) | Detection | $\mathcal{L}(x;\mathcal{M}) \,/\, \mathbb{E}_{\tilde{x}\sim\mathcal{M}}[\mathcal{L}(\tilde{x};\mathcal{M}_{ref})]$ |
| DetectGPT (Mitchell et al., 2023) | Detection | $\mathcal{L}(x;\mathcal{M}) - \mathbb{E}_{\tilde{x}\sim\phi(x)}[\mathcal{L}(\tilde{x};\mathcal{M})]$ |
| Neighborhood (Mattern et al., 2023) | MIA | $\mathcal{L}(x;\mathcal{M}) - \mathbb{E}_{\tilde{x}\sim\phi(x)}[\mathcal{L}(\tilde{x};\mathcal{M})]$ |
| Fast-DetectGPT (Bao et al., 2024) | Detection | $\Phi\left(\mathcal{L}(x;\mathcal{M}) - \mathbb{E}_{\tilde{x}\sim\phi(x)}[\mathcal{L}(\tilde{x};\mathcal{M})]\right)$ |
| Min-$k$% (Shi et al., 2024) | MIA | $\frac{1}{k}\sum_{i\in\min\text{-}k\%}\mathcal{L}(x_i;\mathcal{M})$ |
| Min-$k$%++ (Zhang et al., 2025) | MIA | $\frac{1}{k}\sum_{i\in\min\text{-}k\%}\Phi\left(\mathcal{L}(x_i;\mathcal{M}) - \mathbb{E}_{\tilde{x}_i\sim\mathcal{M}}[\mathcal{L}(\tilde{x}_i;\mathcal{M})]\right)$ |
| Lastde (Xu et al., 2025) | Detection | $\mathcal{L}(x;\mathcal{M}) \,/\, \text{StdDev}(\{\text{DE}(x,\tau)\}_{\tau=1}^{\tau'})$ |
| Lastde++ (Xu et al., 2025) | Detection | $\Phi\left(\text{Lastde}(x) - \mathbb{E}_{\tilde{x}\sim\phi(x)}[\text{Lastde}(\tilde{x})]\right)$ |

Table 1: Unified formulations of membership inference and machine text detection metrics. $\mathcal{L}(x;\mathcal{M})$ denotes negative log likelihood, $\mathcal{R}(x;\mathcal{M})$ denotes average log rank, $\phi(x)$ denotes an arbitrary perturbation function, and $\Phi(x) = \frac{x}{\sigma_x}$ denotes division by the standard deviation. See Appendix A for more details on how we derive each formula.

## 2.5 DISCUSSION

While our general formulation covers many methods from both tasks, there are metrics that fall outside of our core framing—most notably, the metrics that are not ratios but single quantities (e.g., DC-PDD, Min-K%). For these methods, we assess transferability empirically and defer a more fully unified theoretical investigation to future work.

In addition, while most methods exhibit a large degree of transferability, there are some methods whose transferability is less strong. Of particular interest is the zlib method which, despite being an approximate likelihood ratio, performs relatively poorly on generated text detection. We discuss some hypotheses for why this is the case in more detail in Section 3.3.

## 3 MIAs AND MACHINE TEXT DETECTION ARE TRANSFERABLE

### 3.1 EXPERIMENTAL SETUP

**Membership Inference.** We evaluate **MIAs** on the MIMIR dataset (Duan et al., 2024), which is a large-scale MIA benchmark consisting of **5 domains**[2] included in the Pile (Gao et al., 2020): Wikipedia (knowledge), Pile CC (general web), PubMed Central and ArXiv (academic), HackerNews (dialogue). Members and non-members are sampled from the training and test sets of the Pile, respectively and 13-gram filtering is used to ensure no leakage. We target the PYTHIA suite: **5 models** of PYTHIA (Biderman et al., 2023) with 160M, 1.4B, 2.8B, 6.7B, and 12B parameters.

**Machine Text Detection.** To evaluate the performance of the methods on **machine text detection**, we utilize the RAID dataset (Dugan et al., 2024), which is a large-scale detection benchmark consisting of generated text and human-written text in **8 domains**: Wikipedia and News (knowledge), Abstracts (academic), Recipes (instructions), Reddit (dialogue), Poetry (creative), Books (narrative), Reviews (opinions), as well as **5 models**: GPT-2-XL (Radford et al., 2019), MPT-30B-Chat (Team, 2023), LLaMA-2-70B-Chat (Touvron et al., 2023) as open-source models and ChatGPT (OpenAI, 2023) and GPT-4 (OpenAI et al., 2024) as closed-source models.

**Evaluation Measures.** To assess the empirical transferability, we compute the performance ranking of all methods on both tasks using **AUROC** score and report the Spearman's rank correlation

---

[2]To compare the two tasks under a common condition, we focus on textual domains as in machine text detection. See Appendix C for full results on MIAs, including technical domains: GitHub and DM Mathematics.

coefficient to judge how closely the two rankings match each other. A high rank correlation implies that the state-of-the-art metrics for one task will perform at or near state-of-the-art on the other.

**Methods Tested.** For membership inference we consider **7 methods**: *Reference* (Carlini et al., 2021), *Zlib* (Carlini et al., 2021), *Neighborhood attack* (Mattern et al., 2023), *Min-K% Prob* (Shi et al., 2024), *Min-K%++* (Zhang et al., 2025), *ReCaLL* (Xie et al., 2024), and *DC-PDD* (Zhang et al., 2024). For machine text detection, we consider **5 methods**: *DetectGPT* (Mitchell et al., 2023), *Fast-DetectGPT* (Bao et al., 2024), *Binoculars* (Hans et al., 2024), *DetectLLM* (Su et al., 2023), and *Lastde++* (Xu et al., 2025). We also compute the *Loss*, *Rank*, *LogRank*, and *Entropy* as general baseline methods. An overview of the methods tested can be found in Table 1, more details on each method can be found in Appendix A, and the configurations can be found in Appendix B.

**Detection Scenarios.** For our main rank correlation result we target a **white-box setting** for both MIAs and machine text detection, where token probability distributions of target models are accessible. This allows us to compare the two tasks under a common condition and is consistent with our theoretical formulation, which relies on signals derived from target model probabilities.

To further examine transferability in real-world scenarios, we additionally investigate the **black-box setting** for machine text detection, targeting closed-source models such as ChatGPT and GPT-4.[3] In this case, where the target model's logits are not available, we employ surrogate models (PYTHIA-160M (Biderman et al., 2023) and Llama-3-3.2B (Grattafiori et al., 2024)) and report the average detection performance across the surrogates.

## 3.2 MAIN RESULTS

**Substantial Rank Correlation Between MIAs and Machine Text Detection.** Figure 2 illustrates the relationship between the rankings of all methods when evaluated on MIAs and on machine text detection. The rankings are based on average performance on MIAs (across five domains and five target models) and machine text detection (across eight domains and three generators). We compute the rank correlation over all 15 methods and obtain a statistically significant Spearman's correlation of $\rho = 0.66$ ($p < 0.01$). This result indicates that many methods originally proposed for MIAs also perform well in machine text detection, and vice versa. Notably, when focusing on stronger methods (top-10 on MIAs), we observe an even stronger correlation of $\rho = 0.78$ ($p < 0.01$), suggesting strong transferability.

**Superior Performance from the Other Task.** Figure 3 shows the AUROC performance of all methods from both MIAs and machine text detection evaluated on MIAs, and their performance evaluated on machine text detection. The results on MIAs are averaged across five domains and five target models, while those on machine text detection are averaged across eight domains and three generators. Remarkably, we observe that Binoculars, originally proposed as a machine text detector, achieves the best average performance in both MIAs and machine text detection. This result suggests that current evaluations in MIAs may be biased by overlooking stronger methods from machine text detection, potentially leading to conclusions that miss valuable insights. Conversely, in machine text detection, methods from MIAs already demonstrate competitive or even superior performance. These findings call for greater cross-task awareness and development.

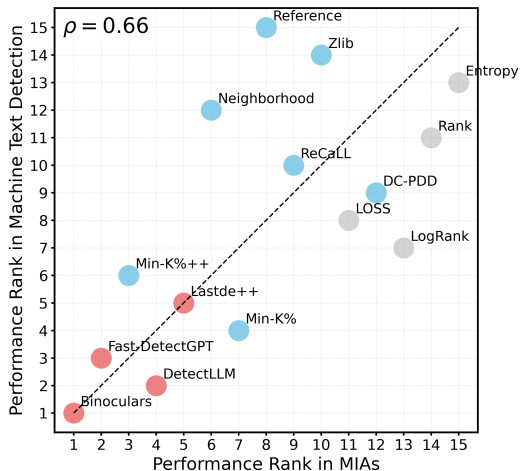

Figure 2: Relationship between method rankings across MIAs and machine text detection. Blue and red plots show MIA methods and machine text detectors. Gray plots indicate general baselines. Dashed line denotes equal ranks.

---

[3]Since the training data of such closed-source models is not accessible, true ground truth for MIAs is inherently not feasible in this setting. Therefore, MIAs are evaluated only in a white-box setting.

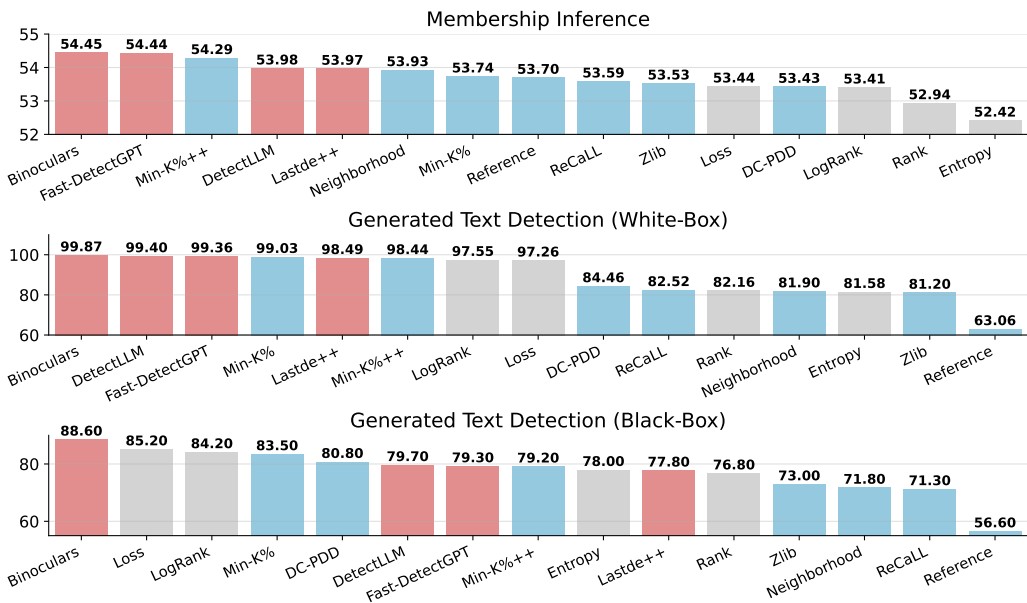

Figure 3: AUROC scores for membership inference attacks (blue bars) and generated text detectors (red bars) across both tasks. **Top**: Membership inference results on the MIMIR benchmark with 13-gram de-duplication filtering averaged over *five domains* and *five models*. **Middle and Bottom**: Generated text detection results on white-box and black-box settings from the RAID benchmark, averaged over *eight domains* and *five models*. Both MIA methods (blue bars) and machine text detectors (red bars) have comparable performance in the cross-task setting. Notably, Binoculars achieves the best average performance in both tasks. Full results are provided in Appendix C.

## 3.3 ANALYSIS

**Transferability in Real-World Scenarios.** We further assess transferability in real-world scenarios by evaluating how well MIA methods perform in a black-box setting of machine text detection on texts generated by ChatGPT and GPT-4. Figure 3 shows the average AUROC performance of all methods from MIAs and machine text detection across eight domains and the two generators. While Binoculars still outperforms other methods by a large margin, MIA methods such as Min-K% and DC-PDD achieve performance on par with or better than strong detectors. These results provide promising evidence of the transferability of MIAs to machine text detection in real-world scenarios.

**Similar Prediction Score Distributions across Methods.** To further illustrate the transferability between MIAs and machine text detection, we compare the prediction score distributions of an MIA method and a machine text detector when applied to the same task. Figure 1 presents the distributions of Min-K%++ and Binoculars, which demonstrate strong cross-task performance, on MIAs and machine text detection. The domain in both tasks is Wikipedia, and the target model or generator is PYTHIA-12B. To quantify the similarity in distributional shape, we compute the Jensen–Shannon distance between the prediction score distributions produced by Min-K%++ and Binoculars within each task: 0.14 for MIAs and 0.11 for machine text detection. These small distances indicate that Min-K%++ and Binoculars produce closely aligned prediction score distributions within both tasks, providing further evidence of their transferability.

**Zlib as an Outlier Illustrates a Task Difference: Different Prior Distributions.** We take Zlib as an example of limited transferability between MIAs and machine text detection. Zlib ranks 10th out of 15 methods in MIAs but drops to 14th in detection, where it calibrates the loss by dividing it by the zlib compression entropy.

In MIAs, both classes are drawn from the same human-written text distribution, whereas this is not the case for machine text detection. Machine-generated texts are known to be more compressible

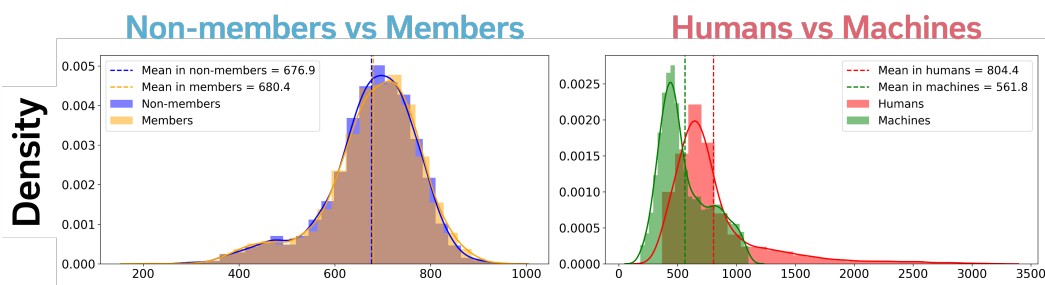

Figure 4: **Zlib compression entropy distribution** in MIAs (non-members vs. members) and machine text detection (humans vs. machines), averaged over 3,000 randomly sampled texts for each dataset. The entropy converges between classes in MIAs but diverges in detection with a long tail, due to different prior distributions.

than human-written ones (Tulchinskii et al., 2023; Mao et al., 2025), and we also find the trend in our setting. As shown in Figure 4, the zlib entropy converges between classes in MIAs but diverges in machine text detection, based on 3,000 randomly sampled texts from all domains and models in each dataset. In the detection, both the loss and the zlib entropy shift in the same direction between human-written and machine-generated texts, yielding similar Zlib scores across classes and little discriminative signal. Consequently, while moderately effective for MIAs, Zlib transfers poorly to machine text detection, highlighting a key task difference: **different prior distributions**. We leave a comprehensive analysis of other methods with limited transferability for future work.

## 4 RELATED WORK

**Proximity between MIAs and Machine Text Detection.** Previous studies have briefly noted the similarity between specific methods in MIAs and machine text detection. For instance, Shi et al. (2024) and Naseh & Mireshghallah (2025) mention that Neighborhood attack (Mattern et al., 2023) from MIAs and DetectGPT (Mitchell et al., 2023) from machine text detection are essentially identical, both of which estimate the probability curvature around a target text by perturbing the text. Most recently, Naseh & Mireshghallah (2025) suggested that MIAs may function as machine text detectors, based on their observations that current MIAs often misclassify machine-generated non-members as members. In contrast to such prior work, we go beyond these limited observations and conduct the first comprehensive study of transferability, introducing a general formulation and validating it with large-scale experiments.

**Optimality of Likelihood Ratio Tests for MIA.** Foundational work by Carlini et al. (2022) was the first to utilize the optimality of the likelihood ratio test as a starting point from which to derive membership inference attacks. They propose an online likelihood ratio attack (LiRA) where they train sets of "shadow" models on random samples of the data distribution with and without the target point $x$. They then fit two gaussians to the confidences of the "in" and "out" models to approximate a parametric likelihood ratio test. They show that these tests achieve strong performance at low false positive rates. While their specific technique is not as practical in the context of multi-billion parameter foundation models with trillion-token pretraining datasets, their work nonetheless provides solid empirical justification for the usage of likelihood ratio tests as a framework from which to reason about strong MIA performance.

**Membership Inference.** Membership inference is a task to determine whether a given sample was part of a given model's training data (Shokri et al., 2017). In the context of language models, recent studies have applied MIAs to pre-training data detection. Since language models tend to show higher likelihood on members compared to non-members, such studies leverage statistical features of a target model, such as likelihood (Carlini et al., 2021), likelihood calibrated by another language model (Carlini et al., 2021), negative log probability curvature (Mattern et al., 2023), and average log-likelihood of the $k\%$ tokens with lowest probabilities (Shi et al., 2024). Other MIA methods are detailed in Appendix A.2.

**Machine-generated Text Detection.** Many studies on machine text detection have investigated supervised methods, including training a classifier with human-written and machine-generated texts with labels. Supervised classifiers aim to capture stylistic or semantic differences between human-written and machine-generated texts, with simple n-gram features (Ippolito et al., 2020; Crothers et al., 2023; Verma et al., 2024) or neural representations (Li et al., 2024; Wang et al., 2024). Since MIAs fundamentally examine how much a target model memorizes a text by exploiting signals from the target model such as likelihood, our investigation of the transferability focuses on zero-shot detectors that similarly rely on statistical features from the model, rather than supervised classifiers that do not. Zero-shot detectors leverage statistical features, including entropy (Lavergne et al., 2008), likelihood (Solaiman et al., 2019), negative curvature of log probabilities (Mitchell et al., 2023), and cross-model entropy (Hans et al., 2024). Other zero-shot detectors are detailed in Appendix A.3.

## 5 CONCLUSION

We comprehensively study the transferability between membership inference attacks (MIAs) and machine-generated text detection. Our theoretical and empirical investigations reveal that 1) Many methods from both tasks that exhibit high transferability can be reduced to a general formulation that measures the discrepancy between the surprisal of a text under a target model and under the true distribution, reflecting their shared objective to approximate the true distribution and contributing to the transferability and 2) Many methods originally proposed for MIAs perform well in machine text detection, and vice versa, as evidenced by substantial rank correlation across the tasks and 3) Notably, a method originally designed for machine text detection surpasses state-of-the-art MIA methods on MIAs, demonstrating the practical impact of the transferability. These findings call for greater cross-task awareness, closer collaboration, and fair evaluation across the two research communities.

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

## A DETAILS OF METHODS

### A.1 BASELINES

(1) **Loss** simply uses the target sample $x$'s loss against the model $\mathcal{M}$: $f(\mathbf{x}; \mathcal{M}) = \mathcal{L}(\mathbf{x}; \mathcal{M})$. The hypothesis is that members and machine-generated texts will have a higher likelihood on average than non-members and human-written texts.

(2) **Entropy** measures the expected likelihood of the next token given the preceding tokens at each time step under the model's distribution. $f(\mathbf{x}; \mathcal{M}) = \mathbb{E}_{\tilde{x} \sim \mathcal{M}}[\mathcal{L}(\tilde{x}; \mathcal{M})]$. The hypothesis is that machine-generated texts and member texts will have low entropy as the model "understands" the context more and can better model the next token distribution.

(3) **Rank** measures the average rank of the next token in the model's probability distribution at each time step. $f(x; \mathcal{M}) = \frac{1}{n} \sum_{i=1}^{n} \mathrm{Rank}(x_i; \mathcal{M})$. The hypothesis is that generated text and member text will have a higher average rank than human-written text.

(4) **LogRank** measures the average log rank of the next token: $f(x; \mathcal{M}) = \frac{1}{n} \sum_{i=1}^{n} \log(\mathrm{Rank}(x_i; \mathcal{M}))$. This metric smooths out contributions from very low rank tokens to reflect the probabilistic nature of rank information. Similarly to the previous metric, the hypothesis is that generated text and member text have a high average log rank.

### A.2 MEMBERSHIP INFERENCE ATTACKS

(1) **Reference** (Carlini et al., 2021) uses the difference in the target sample $x$'s loss between the model $\mathcal{M}$ and another reference model $\mathcal{M}_{ref}$. We follow the original author's implementation by taking a smaller-size reference model for each of the models we tested: $f(\mathbf{x}; \mathcal{M}) = \mathcal{L}(\mathbf{x}; \mathcal{M}) - \mathcal{L}(\mathbf{x}; \mathcal{M}_{ref})$. This method falls into the external reference category of likelihood ratio approximation.

(2) **Zlib** (Carlini et al., 2021) employs the ratio of $\mathcal{L}(\mathbf{x}; \mathcal{M})$ and the zlib compression score of the target sample $x$: $f(\mathbf{x}; \mathcal{M}) = \mathcal{L}(\mathbf{x}; \mathcal{M}) \, / \, \mathrm{zlib}(\mathbf{x})$. The zlib compression score is computed by constructing a dictionary of repeated substrings from the text and encoding each dictionary entry into a

string of bits using Huffman Coding (Huffman, 1952). The compression rate is thus a representation of the entropy of the empirical substring distribution of the text. Thus the zlib method can be equivalently rewritten as $f(\mathbf{x}; \mathcal{M}) = \mathcal{L}(\mathbf{x}; \mathcal{M}) / \mathbb{E}_{x \sim \mathcal{M}_{ref}}[\mathcal{L}(\mathbf{x}; \mathcal{M}_{ref})]$ where $\mathcal{M}_{ref}$ represents that empirical substring distribution. We see here that this method falls into the external reference category of likelihood ratio approximation.

(3) **Neighborhood attack** (Mattern et al., 2023) compares $\mathcal{L}(\mathbf{x}; \mathcal{M})$ to the average loss of *neighborhood samples* $\tilde{\mathbf{x}}$, which are samples crafted by perturbing the target sample $x$: $f(\mathbf{x}; \mathcal{M}) = \mathcal{L}(\mathbf{x}; \mathcal{M}) - \frac{1}{n} \sum_{i=1}^{n} \mathcal{L}(\tilde{\mathbf{x}}^{(i)}; \mathcal{M})$. It hypothesizes that members will show a lower loss compared to their neighborhood samples. Since the quantity $\frac{1}{n} \sum_{i=1}^{n} \mathcal{L}(\tilde{\mathbf{x}}^{(i)}; \mathcal{M}) \approx \mathbb{E}_{x \sim \phi(x)}[\mathcal{L}(\tilde{x}, \mathcal{M})]$ we consider this method to be in the text sampling category of likelihood ratio approximation.

(4) **Min-K%** (Shi et al., 2024) calculates the average of log-likelihood of the $k\%$ tokens with lowest probabilities: $f(\mathbf{x}; \mathcal{M}) = \frac{1}{k} \sum_{x_i \in \text{min-}k(\mathbf{x})} \log p(x_i \mid x_{<i}; \mathcal{M})$. The intuition is that members will include fewer outlier tokens with low probability compared to non-members.

(5) **Min-K%++** (Zhang et al., 2025) computes the average of the log-likelihood of the $k\%$ tokens with lowest probabilities, where each value is standardized over the model's vocabulary: $f(\mathbf{x}; \mathcal{M}) = \frac{1}{k} \sum_{x_i \in \text{min-k}\%} (\log p(x_i \mid x_{<i}; \mathcal{M}) - \mu_{x_{<i}})/\sigma_{x_{<i}}$. In our notation this can be written as:

$$\frac{1}{k} \sum_{x_i \in \text{min-}k\%} \frac{\mathcal{L}(x_i; \mathcal{M}) - \mathbb{E}_{\tilde{x}_i \sim \mathcal{M}} \mathcal{L}(\tilde{x}_i, \mathcal{M})}{\sqrt{\mathbb{E}_{x_i \sim \mathcal{M}}[(\mathcal{L}(x_i; \mathcal{M}) - \mathbb{E}_{\tilde{x}_i \sim \mathcal{M}}[\mathcal{L}(\tilde{x}_i, \mathcal{M})])^2]}}$$

for brevity in our Table 1 we use $\Phi(x) = \frac{x}{\sigma_x}$ as shorthand for the normalization by the standard deviation. Intuitively this metric measures how unexpectedly surprising a particular token is over the vocabulary distribution.

(6) **ReCaLL** (Xie et al., 2024) computes the relative conditional log-likelihood between $x$ and $P \oplus x$ where $P$ is a set of non-member examples $P = p_1 \oplus \cdots \oplus p_n$. It hypothesizes that non-members will have lower log-likelihoods than members, given a non-member context. We represent this in our table as $\mathbb{E}_{\tilde{x} \sim \phi(x)}[\mathcal{L}(\tilde{x}; \mathcal{M})] / \mathcal{L}(x; \mathcal{M})$ where $\phi(x) = P \oplus x$. We consider this method to be in the likelihood ratio by text sampling category.

(7) **DC-PDD** (Zhang et al., 2024) computes the cross-entropy between the token likelihoods under the model $\mathcal{M}$ and the empirical laplace-smoothed unigram token frequency distribution under some reference corpus $\mathcal{D}'$. In the authors' notation this is $f(\mathbf{x}; \mathcal{M}) = -\frac{1}{n} \sum_{i=1}^{n} p(x_i; \mathcal{M}) \cdot \log p(x_i; \mathcal{D}')$ where $p(x_i; \mathcal{D}') = \frac{\text{count}(x_i)+1}{N'+|V|}$. In our table we represent this metric equivalently as $\mathbb{E}_{\tilde{x} \sim \mathcal{M}}[\mathcal{L}(\tilde{x}, \mathcal{M}_{ref})]$ where $\mathcal{M}_{ref}$ represents the unigram token frequency distribution under $\mathcal{D}'$.

### A.3 MACHINE-GENERATED TEXT DETECTORS

(1) **DetectGPT** (Mitchell et al., 2023) computes the degree to which the log-likelihood function under the suspected model has negative curvature for the given target text. They do this by perturbing the sequence using a T5 model Raffel et al. (2023) and evaluating the change in probability. The functional form looks like $f(x; \mathcal{M}) = \log p(x; \mathcal{M}) - \mathbb{E}_{\tilde{x} \sim q(\cdot|x)} \log p(x; \mathcal{M})$. We directly lift this notation for use in our Table 1. This metric is an example of a text sampling based likelihood ratio approximation.

(2) **Fast-DetectGPT** (Bao et al., 2024) forgoes the expensive perturbation approach used by DetectGPT in favor of using the token likelihoods of the perturbation model directly to compute the expectation. They also divide by the mean of sample variances to further smooth out the metric. The full formulation is

$$f(x; \mathcal{M}) = \frac{\log p(x; \mathcal{M}) - \mathbb{E}_{\tilde{x} \sim q(x)}[\log p(\tilde{x}; \mathcal{M})]}{\sqrt{\mathbb{E}_{\tilde{x} \sim q(x)}[(\log p(\tilde{x}; \mathcal{M}) - \mathbb{E}_{\tilde{x} \sim q(x)}[\log p(\tilde{x}; \mathcal{M})])^2]}}$$

We report this using the shorthand $\Phi(x) = \frac{x}{\sigma_x}$ in our Table 1 to represent this quantity. We consider this metric another example of the text sampling based likelihood ratio approach.

(3) **Binoculars** (Hans et al., 2024) computes the ratio of the perplexity to the cross entropy of the text under some reference model $\mathcal{M}_{ref}$. The full formulation using the authors' notation is as follows:

$$B_{\mathcal{M}_1, \mathcal{M}_2}(s) = \frac{\sum_{i=1}^{L} \log\left(\mathcal{M}_1(s)_i\right)}{\sum_{i=1}^{L} \mathcal{M}_1(s)_i \cdot \log\left(\mathcal{M}_2(s)_i\right)}$$

We re-formulate this objective equivalently in our own notation as:

$$\mathcal{L}(x; \mathcal{M}) \,/\, \mathbb{E}_{x \sim \mathcal{M}}[\mathcal{L}(x; \mathcal{M}_{ref})]$$

The intuition behind this metric is that it measures how much more likely a given text is than what we would expect according to some reference model. We consider this metric a case of approximation via reference model.

(4) **DetectLLM** (Su et al., 2023) is a variant of DetectGPT that uses the log rank as the core quantity to test rather than the log likelihood. The authors propose two metrics, Log-Likelihood Log-Rank Ratio (LRR) and Normalized Log-Rank Perturbation (NPR). For the purposes of our work we consider the NPR metric which has superior performance and is characterized by the following formulation:

$$\text{NPR} = \frac{\frac{1}{n}\sum_{p=1}^{n} \log r_\theta(\tilde{x}_p)}{\log r_\theta(x)}$$

where $r_\theta(x_i)$ represents the rank of the token $x$ in the model's output distribution. We reformulate this equivalently as

$$\frac{\mathbb{E}_{\tilde{x} \sim \phi(x)}[\mathcal{R}(\tilde{x}; \mathcal{M})]}{\mathcal{R}(x; \mathcal{M})}$$

using the notation $\mathcal{R}(x; \mathcal{M})$ to denote the log rank similar to how $\mathcal{L}(x; \mathcal{M})$ denotes the log likelihood. While this metric doesn't quite fall cleanly into our likelihood ratio categorization (since it does not compute likelihood) we nonetheless note the strong similarities between this metric and other approximate likelihood ratios.

(5) **Lastde++** (Xu et al., 2025) utilizes a quantity known as multi-scale diversity entropy (MDE) (Wang et al., 2021) to measure the local fluctuations in likelihood across a particular text sequence. Their metric is:

$$\text{Lastde}(x; \mathcal{M}) = \frac{\mathcal{L}(x; \mathcal{M})}{\text{StdDev}(\{\text{DE}(s, \varepsilon, 1), ..., \text{DE}(s, \varepsilon, \tau')\})}$$

$$\text{DE}(s, \varepsilon, \tau) = -\frac{1}{\ln \varepsilon} \sum_{i=1}^{\varepsilon} P_i^{(\tau)} \ln P_i^{(\tau)}$$

Where $P_i^{(\tau)}$ measures the *diversity* of text, i.e. the extent to which adjacent segments of tokens have similar probability sequences (see Xu et al. (2025)). Intuitively the Lastde metric can be thought of as comparing likelihood to the expected diversity. Lastde++ takes this quantity and applies a sampling based perturbation and normalization similar to FastDetectGPT (Bao et al., 2024).

$$\text{Lastde++}(x) = \frac{\text{Lastde}(x; \mathcal{M}) - \mathbb{E}_{\tilde{x} \sim \phi(x)}[\text{Lastde}(\tilde{x}, \mathcal{M})]}{\sqrt{\mathbb{E}_{\tilde{x} \sim \phi(x)}[(\text{Lastde}(\tilde{x}, \mathcal{M}) - \mathbb{E}_{\tilde{x} \sim \phi(x)}[\text{Lastde}(\tilde{x}, \mathcal{M})])^2]}}$$

This can be thought of as measuring whether or not the quantity tracked by $\text{Lastde}(x)$ is at a local maximum for the particular sequence of text.

We consider neither Lastde++ or Lastde to be approximating likelihood ratios as the diversity-entropy (DE) metric seems to be more intuitively thought of as measuring variance rather than likelihood. We leave to future work a more thorough analysis of these variance-based metrics.

## B    MODELING INFORMATION OF METHODS

### B.1    MEMBERSHIP INFERENCE ATTACKS

**Reference**    (Carlini et al., 2021) Following the original author's implementation, we take a smaller-size reference model for each target model. In MIAs, we use PYTHIA-70M for all target models. In

white-box machine text detection, we use GPT2-Small as the smaller model for GPT2-XL, MPT-7B-Chat for MPT-30B-Chat, and LLaMA-7B-Chat for LLaMA-70B-Chat. In black-box machine text detection, we use LLaMA-3-1B as the smaller model for LLaMA-3-3.2B and PYTHIA-70M for PYTHIA-160M.

**Neighborhood attack** (Mattern et al., 2023) We use the repository[4] default settings, namely T5-Large as mask filling model and 0.3 as the masking rate, across MIAs, white-box machine text detection, and black-box machine text detection.

**Min-K%** (Shi et al., 2024) We use the setting that was found to ensure the favorable performance in the original paper, namely $k = 20$ as the percent of tokens with the lowest probabilities. Likewise, the token percentage $k$ for Min-K%++ (Zhang et al., 2025) is also set as $k = 20$ for fair comparison.

**ReCaLL** (Xie et al., 2024) We set the number of prefixes to $n = 10$, which has been shown in the original paper to yield favorable performance. In MIAs, for each domain, we retrieve non-members as prefixes from the Pile dataset, excluding those in the MIMIR benchmark. To adapt ReCaLL to machine text detection, we use human-written texts as prefixes, since they belong to the negative class. For each domain, these human-written prefixes are retrieved from a subset of the RAID benchmark that was not used in our test set.

**DC-PDD** (Zhang et al., 2024) Following the official GitHub repository[5], we take a subset of C4 (Raffel et al., 2023) and build a token frequency distribution with the tokenizer of each target model in MIAs and white-box machine text detection, and of each surrogate model in black-box detection.

## B.2 MACHINE TEXT DETECTORS

**DetectGPT** (Mitchell et al., 2023) We follow the repository[6] default settings, namely T5-Large as mask filling model and 0.3 as the masking rate, across MIAs, white-box machine text detection, and black-box machine text detection.

**Fast-DetectGPT** (Bao et al., 2024) Following the original paper, we employ the target model as both the scoring and perturbation model in MIAs and white-box machine text detection. In black-box detection, we instead use surrogate models as those (see §3.1).

**Binoculars** (Hans et al., 2024) is reported to work best when the target and reference models are similar in performance. Following the original setting, we use the official code[7] to compute perplexity under a reference model. For MIAs, since the PYTHIA does not provide corresponding chat versions for each model size, we use the PYTHIA-deduped models as references. In white-box detection, we use GPT2-XL-Chat [8] for GPT2-XL, MPT-30B for MPT-30B-Chat, and LLaMA-2-70B for LLaMA-2-70B-Chat. In black-box detection, we adopt LLaMA-3.2-3B-instruct for LLaMA-3-3.2B and PYTHIA-160M-deduped for PYTHIA-160M.

**DetectLLM** (Mitchell et al., 2023) In line with Fast-DetectGPT, we utilize the target model as both the scoring and perturbation model in MIAs and white-box detection. For black-box detection, we instead use surrogate models (see §3.1).

**Lastde++** (Xu et al., 2025) In our implementation, we use the official code and use the repository[9] default settings of the sliding window size $s = 4$, the interval precision $\epsilon = 8$, and the number of scales $\tau' = 15$.

---

[4]https://github.com/mireshghallah/neighborhood-curvature-mia
[5]https://github.com/zhang-wei-chao/DC-PDD
[6]https://github.com/eric-mitchell/detect-gpt
[7]https://github.com/ahans30/Binoculars
[8]lgaalves/gpt2-xl_lima
[9]https://github.com/TrustMedia-zju/Lastde_Detector

## C  FULL PERFORMANCE ON MEMBERSHIP INFERENCE AND MACHINE TEXT DETECTION

Figure 2, Figure 3, and 4 report the full results on membership inference, white-box machine text detection, and black-box machine text detection, respectively.

Table 2: Full comparison of membership inference performances (AUROC) of MIA methods and machine text detectors on the MIMIR benchmark with 13-gram deduplication. Target models are PYTHIA with 160M, 1.4B, 2.8B, 6.9B, and 12B parameters. Gray, blue, and red areas indicate general baseline methods, MIA methods, and machine text detectors, respectively. Textual domains are the bolded columns (Wikipedia, Pile CC, PubMed, ArXiv, HackerNews).

| Method | **Wikipedia** | | | | | GitHub | | | | | **Pile CC** | | | | | **PubMed Central** | | | | |
|---|---|---|---|---|---|---|---|---|---|---|---|---|---|---|---|---|---|---|---|---|
| | 160M | 1.4B | 2.8B | 6.9B | 12B | 160M | 1.4B | 2.8B | 6.9B | 12B | 160M | 1.4B | 2.8B | 6.9B | 12B | 160M | 1.4B | 2.8B | 6.9B | 12B |
| Loss | 51.2 | 53.4 | 54.1 | 55.6 | 56.5 | 76.3 | 80.2 | 81.4 | 82.7 | 83.6 | 50.1 | 51.0 | 51.2 | 52.1 | 52.7 | 50.9 | 52.1 | 52.7 | 53.4 | 54.0 |
| Rank | 49.7 | 52.9 | 53.9 | 56.2 | 57.5 | 70.0 | 74.6 | 75.7 | 77.4 | 77.8 | 50.9 | 51.6 | 51.8 | 52.2 | 52.1 | 51.9 | 52.3 | 52.4 | 53.0 | 53.6 |
| LogRank | 51.3 | 53.5 | 54.3 | 55.6 | 56.9 | 75.9 | 80.0 | 81.2 | 82.5 | 83.3 | 50.3 | 51.0 | 51.2 | 52.3 | 52.7 | 50.8 | 51.8 | 52.4 | 53.3 | 53.7 |
| Entropy | 50.9 | 52.1 | 52.6 | 53.1 | 53.4 | 76.3 | 79.7 | 80.7 | 81.9 | 82.7 | 49.6 | 50.2 | 50.5 | 50.9 | 51.3 | 51.7 | 51.8 | 52.1 | 52.2 | 52.3 |
| Reference | 51.7 | 54.4 | 55.2 | 57.4 | 58.5 | 37.3 | 41.0 | 41.8 | 43.2 | 43.6 | 50.9 | 52.7 | 52.8 | 53.9 | 54.5 | 49.4 | 52.2 | 52.6 | 53.5 | 54.1 |
| Zlib | 50.4 | 53.1 | 54.0 | 55.7 | 56.7 | 79.7 | 82.9 | 83.9 | 85.0 | 85.7 | 51.1 | 52.1 | 52.3 | 53.2 | 53.6 | 51.5 | 52.6 | 53.1 | 53.7 | 54.2 |
| Neighborhood | 51.2 | 54.4 | 54.8 | 56.0 | 57.7 | 75.4 | 74.7 | 74.1 | 74.8 | 74.9 | 51.4 | 52.7 | 53.3 | 54.8 | 54.7 | 52.6 | 55.0 | 55.8 | 56.5 | 57.0 |
| Min-K% | 50.6 | 53.5 | 54.7 | 56.7 | 57.8 | 75.2 | 79.8 | 81.0 | 82.5 | 83.4 | 50.7 | 51.4 | 51.6 | 52.5 | 52.9 | 51.4 | 52.6 | 53.0 | 53.9 | 54.8 |
| Min-K%++ | 51.2 | 55.2 | 56.4 | 59.6 | 60.7 | 73.2 | 78.2 | 79.7 | 81.1 | 82.4 | 50.9 | 52.4 | 52.2 | 54.0 | 54.7 | 50.6 | 52.2 | 52.8 | 54.3 | 55.2 |
| ReCaLL | 50.5 | 54.2 | 54.6 | 57.2 | 57.7 | 72.2 | 77.3 | 79.6 | 80.6 | 81.9 | 48.2 | 49.4 | 50.7 | 51.5 | 51.2 | 52.1 | 52.6 | 55.1 | 54.9 | 56.0 |
| DC-PDD | 52.4 | 53.9 | 54.5 | 55.8 | 56.4 | 82.1 | 85.2 | 86.2 | 86.9 | 87.6 | 50.8 | 52.6 | 52.7 | 53.3 | 53.6 | 50.5 | 51.7 | 52.5 | 52.9 | 53.2 |
| DetectGPT | 51.2 | 54.4 | 54.8 | 56.0 | 57.7 | 75.4 | 74.7 | 74.1 | 74.8 | 74.9 | 51.4 | 52.7 | 53.3 | 54.8 | 54.7 | 52.6 | 55.0 | 55.8 | 56.5 | 57.0 |
| Fast-DetectGPT | 51.9 | 54.9 | 56.3 | 60.0 | 62.9 | 57.8 | 67.2 | 69.6 | 71.4 | 72.3 | 51.8 | 54.2 | 53.9 | 55.6 | 56.1 | 49.1 | 51.7 | 52.8 | 55.1 | 56.4 |
| Binoculars | 51.7 | 55.2 | 56.7 | 58.5 | 60.6 | 71.8 | 77.2 | 74.5 | 80.3 | 81.7 | 51.2 | 53.6 | 54.5 | 55.1 | 55.0 | 50.4 | 52.4 | 53.0 | 55.2 | 55.9 |
| DetectLLM | 51.6 | 54.0 | 55.4 | 58.3 | 61.4 | 56.4 | 66.8 | 69.6 | 71.3 | 71.8 | 52.2 | 53.7 | 53.5 | 55.5 | 55.8 | 49.0 | 51.3 | 52.5 | 54.8 | 55.7 |
| Lastde++ | 50.8 | 54.2 | 55.7 | 59.3 | 61.4 | 52.8 | 64.7 | 66.8 | 68.6 | 69.7 | 50.9 | 52.3 | 52.7 | 54.6 | 54.5 | 50.7 | 52.2 | 53.0 | 54.6 | 56.4 |

| Method | **ArXiv** | | | | | DM Mathematics | | | | | **HackerNews** | | | | | Avg. (textual domains) | | | | |
|---|---|---|---|---|---|---|---|---|---|---|---|---|---|---|---|---|---|---|---|---|
| | 160M | 1.4B | 2.8B | 6.9B | 12B | 160M | 1.4B | 2.8B | 6.9B | 12B | 160M | 1.4B | 2.8B | 6.9B | 12B | 160M | 1.4B | 2.8B | 6.9B | 12B |
| Loss | 54.6 | 55.8 | 56.4 | 57.5 | 58.1 | 67.8 | 67.5 | 67.2 | 67.3 | 67.3 | 50.5 | 51.8 | 52.6 | 53.4 | 54.2 | 51.5 | 52.8 | 53.4 | 54.4 | 55.1 |
| Rank | 52.0 | 52.4 | 52.6 | 54.6 | 55.0 | 60.6 | 60.4 | 60.7 | 60.6 | 60.6 | 51.7 | 52.7 | 52.5 | 53.6 | 54.5 | 51.2 | 52.4 | 52.6 | 53.9 | 54.5 |
| LogRank | 54.4 | 55.6 | 56.1 | 57.5 | 57.9 | 66.3 | 66.4 | 66.2 | 66.4 | 66.2 | 50.4 | 51.7 | 52.4 | 53.7 | 54.3 | 51.4 | 52.7 | 53.3 | 54.5 | 55.1 |
| Entropy | 54.8 | 55.4 | 55.1 | 55.7 | 55.7 | 68.4 | 67.6 | 67.4 | 67.2 | 67.1 | 50.5 | 52.2 | 52.0 | 52.1 | 52.1 | 51.5 | 52.3 | 52.5 | 52.8 | 53.0 |
| Reference | 51.2 | 53.3 | 54.1 | 55.8 | 56.8 | 45.0 | 44.8 | 44.4 | 44.4 | 44.3 | 50.0 | 52.1 | 53.8 | 55.2 | 56.6 | 50.6 | 52.9 | 53.7 | 55.2 | 56.1 |
| Zlib | 54.2 | 55.3 | 55.7 | 56.7 | 57.2 | 64.6 | 64.7 | 64.6 | 64.6 | 64.6 | 51.2 | 51.9 | 52.4 | 52.9 | 53.4 | 51.7 | 53.0 | 53.5 | 54.4 | 55.0 |
| Neighborhood | 53.1 | 54.7 | 54.2 | 54.9 | 55.0 | 53.3 | 51.8 | 53.1 | 50.8 | 53.3 | 51.1 | 51.0 | 51.8 | 51.9 | 52.6 | 51.0 | 52.2 | 52.6 | 53.4 | 53.9 |
| Min-K% | 53.3 | 55.0 | 55.8 | 57.3 | 58.4 | 64.7 | 65.2 | 64.9 | 65.1 | 65.1 | 50.9 | 51.8 | 53.1 | 54.3 | 55.3 | 51.4 | 52.9 | 53.6 | 54.9 | 55.8 |
| Min-K%++ | 51.3 | 53.8 | 55.9 | 56.9 | 59.9 | 58.8 | 57.9 | 58.7 | 58.4 | 58.2 | 51.1 | 51.5 | 53.1 | 54.6 | 56.4 | 51.0 | 53.0 | 54.1 | 55.9 | 57.4 |
| ReCaLL | 53.4 | 54.5 | 55.4 | 57.3 | 58.3 | 58.0 | 56.4 | 56.8 | 53.7 | 53.1 | 52.6 | 52.4 | 52.4 | 53.7 | 54.0 | 51.4 | 52.6 | 53.6 | 54.9 | 55.4 |
| DC-PDD | 54.7 | 56.3 | 56.3 | 57.4 | 57.7 | 63.9 | 63.8 | 63.5 | 63.4 | 63.6 | 49.4 | 51.0 | 51.4 | 52.1 | 52.7 | 51.6 | 53.1 | 53.5 | 54.3 | 54.7 |
| DetectGPT | 53.1 | 54.7 | 54.2 | 54.9 | 55.0 | 53.3 | 51.8 | 53.1 | 50.8 | 53.3 | 51.1 | 51.0 | 51.8 | 51.9 | 52.6 | 51.0 | 52.2 | 52.6 | 53.4 | 53.9 |
| Fast-DetectGPT | 51.5 | 53.2 | 54.9 | 57.4 | 59.3 | 52.4 | 53.1 | 54.2 | 54.0 | 54.7 | 50.1 | 49.5 | 51.9 | 54.1 | 56.4 | 50.9 | 52.7 | 54.0 | 56.4 | 58.2 |
| Binoculars | 54.1 | 54.6 | 54.9 | 57.6 | 59.9 | 55.7 | 53.8 | 53.5 | 52.1 | 52.8 | 49.6 | 50.4 | 51.5 | 53.8 | 55.6 | 51.3 | 53.3 | 54.2 | 56.1 | 57.5 |
| DetectLLM | 51.5 | 53.1 | 54.7 | 57.6 | 58.6 | 51.9 | 51.8 | 53.2 | 53.6 | 53.0 | 49.8 | 49.2 | 51.1 | 53.6 | 55.5 | 50.8 | 52.3 | 53.4 | 56.0 | 57.4 |
| Lastde++ | 50.8 | 52.9 | 54.7 | 56.1 | 58.1 | 52.3 | 51.6 | 52.1 | 52.2 | 52.3 | 50.0 | 51.1 | 52.0 | 54.0 | 56.0 | 50.6 | 52.5 | 53.6 | 55.7 | 57.3 |

Table 3: Full comparison of white-box machine text detection performances (AUROC) of MIA methods and machine text detectors on the RAID benchmark. Target models are GPT-2: GPT-2-XL, MPT: MPT-30B-Chat, LLaMA: LLaMA-2-70B-Chat. Gray, blue, and red areas indicate general baseline methods, MIA methods, and machine text detectors, respectively.

| Method | Abstracts | | | Books | | | News | | | Poetry | | |
|---|---|---|---|---|---|---|---|---|---|---|---|---|
| | GPT-2 | MPT | LLaMA | GPT-2 | MPT | LLaMA | GPT-2 | MPT | LLaMA | GPT-2 | MPT | LLaMA |
| Loss | 97.4 | 98.6 | 100.0 | 98.6 | 99.9 | 100.0 | 94.6 | 99.9 | 100.0 | 96.8 | 98.3 | 96.2 |
| Rank | 98.2 | 53.9 | 93.8 | 99.4 | 91.9 | 96.5 | 95.5 | 70.3 | 86.1 | 98.0 | 85.6 | 88.8 |
| LogRank | 98.5 | 96.4 | 100.0 | 99.4 | 99.9 | 100.0 | 96.5 | 99.7 | 100.0 | 97.6 | 98.3 | 95.6 |
| Entropy | 61.6 | 55.0 | 99.9 | 60.7 | 97.4 | 99.9 | 38.9 | 92.3 | 99.9 | 89.3 | 90.3 | 95.4 |
| Reference | 21.7 | 52.4 | 53.5 | 59.7 | 81.7 | 65.2 | 45.9 | 97.8 | 81.4 | 40.8 | 88.7 | 73.6 |
| Zlib | 87.1 | 47.0 | 99.9 | 64.1 | 66.8 | 96.0 | 55.8 | 80.4 | 99.7 | 73.4 | 73.9 | 87.6 |
| Neighborhood | 95.3 | 58.4 | 94.7 | 92.1 | 64.8 | 86.3 | 85.3 | 67.7 | 88.9 | 83.0 | 73.2 | 76.2 |
| Min-K% | 99.8 | 98.8 | 100.0 | 99.9 | 99.9 | 100.0 | 99.4 | 99.9 | 100.0 | 99.3 | 99.0 | 96.8 |
| Min-K%++ | 99.6 | 99.6 | 97.1 | 99.8 | 99.0 | 98.2 | 99.7 | 99.5 | 98.9 | 99.4 | 99.1 | 90.3 |
| ReCaLL | 62.6 | 87.1 | 99.6 | 81.1 | 95.3 | 97.8 | 68.8 | 89.9 | 93.3 | 76.9 | 87.7 | 97.6 |
| DC-PDD | 48.6 | 95.3 | 100.0 | 64.1 | 99.8 | 99.8 | 47.4 | 99.3 | 99.9 | 82.5 | 97.0 | 94.2 |
| DetectGPT | 95.3 | 58.4 | 94.7 | 92.1 | 64.8 | 86.3 | 85.3 | 67.7 | 88.9 | 83.0 | 73.2 | 76.2 |
| Fast-DetectGPT | 99.7 | 100.0 | 99.8 | 99.8 | 100.0 | 99.9 | 99.7 | 100.0 | 100.0 | 99.3 | 100.0 | 94.4 |
| Binoculars | 99.8 | 99.9 | 100.0 | 99.8 | 100.0 | 100.0 | 99.8 | 99.9 | 100.0 | 99.5 | 100.0 | 100.0 |
| DetectLLM | 99.7 | 100.0 | 100.0 | 99.8 | 100.0 | 99.8 | 99.7 | 100.0 | 100.0 | 99.4 | 99.9 | 93.2 |
| Lastde++ | 99.7 | 100.0 | 97.8 | 99.8 | 100.0 | 98.8 | 99.6 | 100.0 | 100.0 | 99.4 | 99.9 | 88.8 |

| Method | Recipes | | | Reddit | | | Reviews | | | Wikipedia | | |
|---|---|---|---|---|---|---|---|---|---|---|---|---|
| | GPT-2 | MPT | LLaMA | GPT-2 | MPT | LLaMA | GPT-2 | MPT | LLaMA | GPT-2 | MPT | LLaMA |
| Loss | 70.4 | 100.0 | 100.0 | 97.7 | 97.4 | 99.7 | 97.9 | 99.9 | 99.9 | 91.4 | 100.0 | 99.9 |
| Rank | 82.3 | 48.8 | 17.5 | 98.3 | 73.9 | 64.6 | 98.8 | 82.8 | 95.3 | 97.0 | 73.4 | 81.3 |
| LogRank | 71.5 | 99.9 | 100.0 | 98.3 | 96.2 | 99.7 | 98.8 | 99.9 | 99.9 | 95.5 | 99.9 | 99.8 |
| Entropy | 40.2 | 98.4 | 100.0 | 72.3 | 78.3 | 99.7 | 59.0 | 98.6 | 99.9 | 37.6 | 94.5 | 99.1 |
| Reference | 34.9 | 84.7 | 12.6 | 51.8 | 53.2 | 85.4 | 60.4 | 78.0 | 85.9 | 46.8 | 92.6 | 64.9 |
| Zlib | 61.0 | 98.0 | 99.9 | 96.0 | 50.9 | 99.6 | 60.3 | 65.9 | 99.2 | 87.4 | 99.3 | 99.8 |
| Neighborhood | 82.5 | 58.2 | 89.2 | 97.5 | 57.0 | 92.1 | 90.6 | 70.8 | 97.2 | 93.7 | 82.4 | 88.6 |
| Min-K% | 89.4 | 100.0 | 100.0 | 99.3 | 96.6 | 99.7 | 99.7 | 99.9 | 100.0 | 99.4 | 100.0 | 99.9 |
| Min-K%++ | 98.9 | 98.8 | 99.2 | 99.3 | 96.9 | 91.7 | 99.4 | 99.7 | 99.5 | 99.6 | 99.6 | 99.7 |
| ReCaLL | 60.2 | 97.1 | 100.0 | 40.5 | 87.8 | 98.6 | 67.4 | 89.4 | 95.8 | 37.0 | 71.1 | 97.9 |
| DC-PDD | 41.3 | 99.9 | 100.0 | 62.9 | 97.7 | 99.9 | 62.2 | 99.8 | 99.9 | 38.3 | 98.3 | 98.9 |
| DetectGPT | 82.5 | 58.2 | 89.2 | 97.5 | 57.0 | 92.1 | 90.6 | 70.8 | 97.2 | 93.7 | 82.4 | 88.6 |
| Fast-DetectGPT | 99.1 | 100.0 | 100.0 | 99.8 | 99.9 | 94.5 | 99.6 | 99.8 | 99.6 | 99.8 | 100.0 | 100.0 |
| Binoculars | 99.6 | 100.0 | 100.0 | 99.7 | 99.1 | 100.0 | 99.8 | 99.9 | 100.0 | 99.8 | 100.0 | 100.0 |
| DetectLLM | 99.0 | 100.0 | 99.9 | 99.8 | 99.9 | 96.7 | 99.7 | 99.8 | 99.7 | 99.8 | 100.0 | 99.9 |
| Lastde++ | 99.0 | 100.0 | 99.4 | 99.8 | 99.6 | 85.1 | 99.5 | 99.6 | 98.0 | 99.9 | 100.0 | 99.9 |

Table 4: Full comparison of black-box machine text detection performances (AUROC) of MIA methods and machine text detectors on the RAID benchmark. Target models are ChatGPT and GPT-4. Surrogate models are Llama: LLaMA-3-3.2B, Pythia: PYTHIA-160M. Gray, blue, and red areas indicate general baseline methods, MIA methods, and machine text detectors, respectively.

| Method | Abstracts ChatGPT Llama | Abstracts ChatGPT Pythia | Abstracts GPT-4 Llama | Abstracts GPT-4 Pythia | Books ChatGPT Llama | Books ChatGPT Pythia | Books GPT-4 Llama | Books GPT-4 Pythia | News ChatGPT Llama | News ChatGPT Pythia | News GPT-4 Llama | News GPT-4 Pythia | Poetry ChatGPT Llama | Poetry ChatGPT Pythia | Poetry GPT-4 Llama | Poetry GPT-4 Pythia |
|---|---|---|---|---|---|---|---|---|---|---|---|---|---|---|---|---|
| Loss | 96.0 | 95.6 | 38.5 | 51.9 | 99.8 | 98.2 | 79.1 | 72.0 | 99.5 | 99.1 | 66.1 | 84.0 | 88.9 | 79.4 | 46.0 | 51.0 |
| Rank | 73.6 | 82.9 | 49.1 | 57.3 | 99.5 | 91.8 | 96.0 | 59.3 | 93.2 | 92.3 | 62.6 | 69.2 | 85.6 | 73.9 | 48.0 | 36.5 |
| LogRank | 96.6 | 96.2 | 40.3 | 53.8 | 99.8 | 97.5 | 77.6 | 65.1 | 99.7 | 98.8 | 68.1 | 80.4 | 88.9 | 76.1 | 44.1 | 43.0 |
| Entropy | 74.5 | 68.3 | 19.8 | 39.2 | 99.4 | 86.2 | 83.7 | 62.3 | 99.1 | 87.4 | 68.0 | 69.3 | 87.4 | 62.3 | 44.7 | 37.3 |
| Reference | 37.6 | 50.3 | 29.0 | 38.9 | 58.4 | 89.2 | 50.6 | 70.4 | 50.0 | 54.8 | 33.8 | 42.1 | 79.5 | 77.2 | 71.4 | 50.1 |
| Zlib | 53.3 | 39.7 | 12.4 | 14.8 | 72.5 | 59.8 | 64.1 | 62.2 | 89.6 | 76.9 | 70.5 | 72.7 | 68.2 | 56.0 | 60.2 | 63.4 |
| Neighborhood | 61.0 | 77.1 | 29.4 | 45.9 | 78.9 | 87.4 | 57.4 | 58.8 | 81.6 | 88.3 | 53.9 | 61.4 | 74.5 | 78.0 | 66.7 | 61.8 |
| Min-K% | 98.3 | 97.5 | 56.1 | 62.5 | 99.8 | 96.4 | 74.4 | 58.6 | 99.6 | 98.5 | 63.5 | 76.3 | 89.8 | 77.0 | 42.3 | 37.2 |
| Min-K%++ | 99.1 | 99.3 | 71.9 | 68.7 | 90.3 | 98.9 | 36.7 | 70.4 | 73.6 | 99.7 | 38.0 | 86.3 | 65.5 | 92.4 | 44.4 | 69.0 |
| ReCaLL | 99.3 | 98.0 | 81.1 | 97.4 | 96.3 | 87.6 | 65.6 | 83.2 | 87.4 | 66.5 | 47.0 | 55.7 | 89.0 | 73.8 | 52.3 | 54.5 |
| DC-PDD | 88.7 | 85.9 | 38.5 | 62.5 | 99.0 | 96.1 | 77.2 | 86.0 | 93.6 | 85.1 | 47.6 | 72.0 | 87.2 | 80.9 | 51.0 | 64.9 |
| DetectGPT | 61.0 | 77.1 | 29.4 | 45.9 | 78.9 | 87.4 | 57.4 | 58.8 | 81.6 | 88.3 | 53.9 | 61.4 | 74.5 | 78.0 | 66.7 | 61.8 |
| Fast-DetectGPT | 99.3 | 98.5 | 77.0 | 58.7 | 83.9 | 96.1 | 38.0 | 69.9 | 72.2 | 99.4 | 46.5 | 88.0 | 70.9 | 89.7 | 53.2 | 79.0 |
| Binoculars | 100.0 | 98.1 | 96.5 | 94.5 | 99.4 | 97.1 | 77.0 | 70.5 | 98.8 | 78.6 | 83.0 | 51.2 | 98.9 | 85.9 | 57.4 | 73.0 |
| DetectLLM | 98.9 | 98.0 | 77.1 | 57.7 | 85.6 | 94.9 | 42.7 | 66.7 | 77.8 | 99.1 | 51.0 | 85.2 | 75.7 | 88.6 | 56.0 | 74.8 |
| Lastde++ | 98.6 | 96.9 | 75.8 | 58.3 | 83.6 | 91.8 | 40.2 | 60.6 | 69.1 | 98.6 | 45.2 | 84.2 | 71.9 | 84.7 | 53.0 | 67.0 |

| Method | Recipes ChatGPT Llama | Recipes ChatGPT Pythia | Recipes GPT-4 Llama | Recipes GPT-4 Pythia | Reddit ChatGPT Llama | Reddit ChatGPT Pythia | Reddit GPT-4 Llama | Reddit GPT-4 Pythia | Reviews ChatGPT Llama | Reviews ChatGPT Pythia | Reviews GPT-4 Llama | Reviews GPT-4 Pythia | Wikipedia ChatGPT Llama | Wikipedia ChatGPT Pythia | Wikipedia GPT-4 Llama | Wikipedia GPT-4 Pythia |
|---|---|---|---|---|---|---|---|---|---|---|---|---|---|---|---|---|
| Loss | 99.7 | 99.0 | 93.4 | 94.8 | 98.8 | 96.6 | 91.6 | 87.0 | 99.9 | 99.5 | 89.0 | 80.9 | 99.0 | 97.1 | 73.8 | 82.4 |
| Rank | 56.9 | 89.7 | 48.0 | 90.5 | 87.7 | 86.2 | 79.0 | 75.7 | 99.1 | 94.7 | 75.7 | 63.1 | 94.8 | 93.3 | 77.2 | 75.4 |
| LogRank | 99.5 | 98.9 | 92.9 | 94.6 | 98.5 | 95.6 | 90.4 | 84.3 | 99.9 | 99.2 | 86.2 | 72.2 | 99.2 | 97.4 | 76.0 | 82.3 |
| Entropy | 99.7 | 92.6 | 93.5 | 83.1 | 96.5 | 88.5 | 90.8 | 82.1 | 99.9 | 93.4 | 91.2 | 66.0 | 98.7 | 83.4 | 73.6 | 72.9 |
| Reference | 55.6 | 36.5 | 36.5 | 29.4 | 60.5 | 83.3 | 52.4 | 74.6 | 76.8 | 94.8 | 58.8 | 82.4 | 52.4 | 50.0 | 41.3 | 42.9 |
| Zlib | 96.9 | 93.4 | 90.9 | 90.8 | 78.0 | 64.9 | 94.0 | 92.8 | 91.2 | 75.8 | 68.5 | 65.3 | 99.9 | 99.9 | 98.0 | 98.6 |
| Neighborhood | 66.3 | 87.5 | 58.6 | 78.8 | 76.4 | 86.6 | 74.6 | 72.1 | 94.3 | 87.6 | 72.7 | 55.3 | 92.9 | 91.7 | 71.7 | 68.0 |
| Min-K% | 99.5 | 98.7 | 92.8 | 93.9 | 98.3 | 93.9 | 87.5 | 79.6 | 99.9 | 98.1 | 79.5 | 65.4 | 99.5 | 97.9 | 78.9 | 81.2 |
| Min-K%++ | 95.1 | 99.7 | 76.6 | 94.5 | 89.0 | 96.8 | 64.8 | 78.6 | 91.5 | 99.2 | 46.0 | 81.6 | 84.5 | 99.3 | 52.3 | 80.8 |
| ReCaLL | 95.9 | 88.0 | 69.2 | 74.2 | 97.0 | 77.3 | 58.2 | 43.2 | 87.9 | 65.0 | 58.4 | 67.7 | 80.3 | 28.0 | 34.6 | 22.9 |
| DC-PDD | 98.2 | 94.4 | 88.1 | 84.7 | 98.6 | 96.0 | 87.3 | 86.2 | 99.8 | 97.0 | 86.7 | 90.8 | 88.3 | 71.9 | 45.2 | 55.2 |
| DetectGPT | 66.3 | 87.5 | 58.6 | 78.8 | 76.4 | 86.6 | 74.6 | 72.1 | 94.3 | 87.6 | 72.7 | 55.3 | 92.9 | 91.7 | 71.7 | 68.0 |
| Fast-DetectGPT | 84.1 | 99.8 | 67.2 | 98.8 | 85.8 | 94.1 | 66.3 | 85.0 | 81.2 | 99.8 | 40.4 | 86.6 | 79.8 | 99.9 | 54.9 | 93.0 |
| Binoculars | 99.9 | 98.2 | 97.8 | 94.7 | 99.6 | 91.3 | 94.6 | 76.1 | 99.5 | 96.5 | 84.0 | 83.8 | 98.8 | 97.7 | 85.1 | 77.8 |
| DetectLLM | 82.2 | 99.8 | 66.6 | 99.1 | 85.9 | 92.6 | 67.4 | 83.6 | 84.6 | 99.7 | 42.2 | 84.8 | 82.1 | 99.9 | 56.8 | 92.6 |
| Lastde++ | 83.9 | 99.4 | 67.8 | 97.9 | 83.9 | 90.4 | 66.6 | 83.9 | 84.2 | 99.3 | 40.9 | 80.4 | 81.6 | 99.8 | 57.4 | 92.0 |

