# OpenReview forum: "Machine Text Detectors are Membership Inference Attacks"
_ICLR.cc/2026/Conference — Submitted to ICLR 2026_

### Official Review · Reviewer_qgWS · 2025-10-16

**Soundness:** 3
**Presentation:** 3
**Contribution:** 3
**Rating:** 6
**Confidence:** 2

**Summary:**

This paper shows that membership inference attacks (MIAs) and machine-generated text detection (MGTD) are two faces of the same statistical problem: under standard asymptotic conditions, both are optimally solved by the same likelihood-ratio test comparing a target language model’s distribution to the true human-text distribution, which the authors prove is uniformly most powerful and whose maximum advantage is bounded via Pinsker’s inequality; they then recast many popular metrics—e.g., DetectGPT/Neighborhood (perturbation curvature), Binoculars (cross-model entropy), Min-K% variants, Reference/Zlib/DC-PDD—explicitly as approximations to this optimal statistic. Building on this unified view, the paper conducts large-scale experiments covering 7 state-of-the-art MIAs and 5 MGTDs across 13 domains and 10 generators, finding strong cross-task transferability (Spearman’s ρ≈0.66 overall, ρ≈0.78 for stronger methods) and, strikingly, that the MGTD method Binoculars attains state-of-the-art performance on MIAs as well; they also analyze outliers such as Zlib, whose poor transfer is attributed to differing priors (machine text being more compressible), and release MINT, a unified evaluation suite to facilitate fair, cross-task comparisons.

**Strengths:**

1. This paper is the first to connect machine-generated text (MGT) detection with membership inference attacks (MIA), constituting a powerful theoretical contribution.
2. Extensive empirical studies substantiate the conclusions.
3. The figures and tables are clear and provide strong support for the claims.

**Weaknesses:**

1. Can your theory be extended to multilingual settings?
2. In **Theorem 2.3**, you state that all machine-generated text detection algorithms are effectively optimizing the single score $\frac{L(x; M)}{L(x; Q)}$, and since the true human distribution $L(x; Q)$ is not directly accessible, perturbation-based methods (e.g., DetectGPT) approximate $L(x; Q)$ via multiple random perturbations. Therefore, the more perturbations that are performed—or the more perturbed segments that are generated—the higher the detection accuracy, correct? In the original DetectGPT paper, it seems only the claim that “generating more perturbed segments yields higher accuracy” is established; I am curious whether the other claim (i.e., increasing the number of perturbation rounds also improves accuracy) holds as well.
3. I am very curious how certain attacks on MGT detectors affect your proposed optimal statistic—for example, the attack in paper [a]. When discussing attacks, we often assume that making LLM-generated text more human-like will better fool detectors; however, some attacks do not behave exactly this way. This paper [a] finds an interesting fact that is, if you use **I**n-**C**ontext **L**earning (ICL) to generate more human-like texts, the detectors tend to be more accurate. It would be great if the paper could include some discussion along these lines.

[a] Wang, Yichen, et al. "Stumbling Blocks: Stress Testing the Robustness of Machine-Generated Text Detectors Under Attacks." Proceedings of the 62nd Annual Meeting of the Association for Computational Linguistics (Volume 1: Long Papers). 2024.

**Questions:**

See weaknesses. I'm not **very sure** about my evaluation (especially the central part of MIA), and I will update (most likely raise) my ratings and confidence after rebuttal if the authors can address my concerns.

---

> ### Author Response · Authors · 2025-11-26
> **Rebuttal by Authors**
>
> Thank you for the valuable feedback. We truly appreciate the time and effort the reviewer invested in evaluating our work. We address the concerns below.
>
> ---
>
> ### **Response to Weakness 1: Applicability of the Theory to Multilingual Settings**
>
> Yes! There is nothing language specific about our theory.
>
> ---
>
> ### **Response to Weakness 2: Clarification of the Effect of Perturbation Segments and Rounds**
>
> We view DetectGPT as performing an approximate marginalization of text likelihood over semantically equivalent perturbed text segments. For this reason, **increasing the number of perturbed segments** is equivalent to increasing the sample size of the approximation **resulting in a more accurate** score. **Increasing the number of perturbation rounds** however, **may not always improve the approximation**. Take, for example, a case where repeated perturbations **destroy the semantics of the underlying text**. Such a case would result in the overall perturbation function no longer sampling semantically equivalent texts and would likely not improve the accuracy of the method.
>
> ---
>
> ### **Response to Weakness 3: Impact of Adversarial Attacks on the Optimal Statistic**
>
> In general, **it should always be the case that perturbations that make text more "human-like"** (e.g. make the $x$ more likely under $Q$ than $M$) **reduce the performance of a detector**. To formalize this notion, let $\phi(x) \rightarrow \tilde{x}$ be some perturbation such that the likelihood of $x$ under $Q$ is increased in expectation $\mathbb{E}_{\tilde{x}\sim\phi(x)}[L(\tilde{x};Q)] \geq L(x;Q)$
>
> and the likelihood under $M$ is decreased in expectation $\mathbb{E}_{\tilde{x}\sim\phi(x)}[L(\tilde{x};M)] \leq L(x;M)$.
>
> **Such a perturbation should always reduce the expected accuracy of the optimal detector**
> $\mathbb{E}_{\tilde{x}\sim\phi(x)}[\frac{L(\tilde{x};M)}{L(\tilde{x};Q)} ]\leq \frac{L(x;M)}{L(x;Q)}$.
>
> After reading through paper [1] we were **unable to find evidence of the ICL attack making detectors more accurate**. In Table 5 of paper [1] **all detectors except SimpleAI had *reduced* performance when the ICL attack** was applied—which is consistent with our understanding of the theory. Please let us know if we're understanding your comment correctly or if there is something we’ve missed here.
>
> ---
>
> References:
>
> [1] Wang et al. Stumbling Blocks: Stress Testing the Robustness of Machine-Generated Text Detectors Under Attacks. ACL 2024.

---

> > ### Comment · Reviewer_qgWS · 2025-11-26
> >
> > Regarding the third weaknesses I mentioned earlier, I realized that I had misremembered part of the paper. My apologies—let me correct it here. As shown in Table 5 of paper [a], ICL has almost no impact on the detector’s accuracy (I do not think the reported 95% AUROC indicates a degradation in detection ability). Could you explain why this phenomenon occurs?

---

> > ### Comment · Reviewer_qgWS · 2025-11-26
> >
> > Regarding your response to the second weakness, I believe my understanding is now correct. I have increased my confidence.

---

> ### Author Response · Authors · 2025-11-26
> **Response to Reviewer**
>
> Thank you for the correction. We acknowledge that the impact of ICL in that paper is minimal. However, we maintain that generating human-like text reduces detector performance (as supported by our formalization). We would attribute the limited impact in their study to their 1-shot setting (using only a single example pair), which was likely insufficient to effectively steer the LLM to generate human-like text.
>
> We believe that a stronger few-shot approach would better minimize the distributional gap and thus degrade detection performance, aligning with our formalization in Response to Weakness 3.

---

> > ### Comment · Reviewer_qgWS · 2025-11-26
> >
> > Thank you for your further clarify. Can you list more paper to support your claim *"We would attribute the limited impact in their study to their 1-shot setting (using only a single example pair), which was likely insufficient to effectively steer the LLM to generate human-like text."*? One example that comes to mind is OUTFOX, where the authors use adversarial training to generate samples that appear more human-like. Could you provide a few additional examples?

---

> > > ### Author Response · Authors · 2025-11-26
> > >
> > > Sure - OUTFOX is indeed a great example of this: they use in-context learning with 5 human-written and 5 machine-generated text examples to generate more human-like texts, leading to a substantial degradation in detection performance.
> > >
> > > Here is another great example paper: "_How Well Do LLMs Imitate Human Writing Style?_" (arxiv.org/abs/2509.24930), which shows that few-shot prompting yields substantially higher human-style alignment compared to zero-shot or one-shot prompting. This supports our interpretation that a single example pair is often insufficient to reliably steer the LLM toward human-like style.
> > >
> > > We hope these examples help clarify our point.

---

> > > > ### Comment · Reviewer_qgWS · 2025-11-26
> > > >
> > > > Thank you for the response. I now have a deeper understanding of the theoretical aspects of the paper. I will keep my current acceptance rating unchanged.

---

> > > > > ### Author Response · Authors · 2025-11-28
> > > > >
> > > > > Thank you for letting us know, and we appreciate your time and assessment!
> > > > > If you have any additional questions or would like further clarification on any aspect of the work, we would be more than happy to discuss them.

---

### Official Review · Reviewer_EPu5 · 2025-10-22

**Soundness:** 2
**Presentation:** 3
**Contribution:** 2
**Rating:** 2
**Confidence:** 4

**Summary:**

This is an interesting paper which seeks to tie together the ideas of detection of machine generated text and of membership inference attacks. I think more of this type of paper should exist - as our field expands exponentially it is really useful to have analysis which tries to tie things together. This isn't necessarily a new idea, the authors cite two articles noting the similarity between 'Membership attack' and 'DetectGPT', each a fundamental idea in its own domain. The authors of this paper seek to build upon this by quantifying the transferability of techniques from one problem (detecting machine generated text) to the other (membership inference) using a range of strategies.

Given that the fundamental observation that membership attack and machine generated text detection may rely on similar methods isn't new, I think there's a fairly high bar to pass in terms of the quality of the analysis. I raise some concerns below about the robustness of the comparison to different methods of generating machine generated text and about the breadth of the analysis.

**Strengths:**

There is a useful (and I think new) conclusion that Binoculars (developed for machine generated text detection) appears to outperform all of the specialised techniques for membership inference attacks on the problem of membership inference.

The writing is very clear, I am not very familiar with the literature on membership inference attacks but had no problem understanding the points that the authors were making.

**Weaknesses:**

Essentially all of the complaints below boil down to not feeling that the analysis is sufficiently in depth.

1) The choice of baselines for machine generated text detection appears incomplete, and nearly all are focused on the same essential strategy (comparing how likely a machine finds the text, either through log-likelihood or log-rank). I believe all of the techniques apart from Lastde take log-likelihood as their essential measure. What about other strategies taking a fundamentally different approach? e.g. those seeking to gain information from fluctuations in the log-likelihood, e.g. (Yang Xu, Yu Wang, Hao An, Zhichen Liu, and Yongyuan Li. 2024. Detecting Subtle Differences between Human and Model Languages Using Spectrum of Relative Likelihood. In Proceedings of the 2024 Conference on Empirical Methods in Natural Language Processing, pages 10108–10121, Miami, Florida, USA. Association for Computational Linguistics.) Or the intrinsic dimension approach Tulchinskii, E., Kuznetsov, K., Kushnareva, L., Cher-niavskii, D., Nikolenko, S., Burnaev, E., Baran-nikov, S., and Piontkovskaya, I. (2024). Intrinsic Dimension Estimation for Robust Detection of AI-Generated Texts. Advances in Neural Information Processing Systems, 36.

2) I'm not convinced by the way that figure 2 is plotted. Why does it make sense to plot performance rank against performance rank, rather than performance against performance?

3) I find it hard to draw strong conclusions from Figure 3, when the numbers are so close. I don't think it's good that the experiments are run in a setting where all detectors perform so well, do we have enough data to conclude that AUROCs of 99.4 and 98.4 are meaningfully different?

4) Beyond this, I don't think the experiment run with the three charts plotted in Figure 3 is particularly useful to evaluate the relative performances on different tasks. Should I think that the top and middle parts of Figure 3 are very similar or not? Logrank and Loss seem to have wildly different performances in the two settings. I think I would agree with the papers conclusion, 'These results provide
promising evidence of the transferability of MIAs to machine text detection in real-world scenarios.', but I had hoped for more than promising evidence and instead a thorough and comprehensive analysis.

5) There is a general trend within papers looking at detection of machine generated text (and ML more widely) for mutually contradictory rankings to appear in different papers. I think it's incumbent upon all of us writing in this domain to carefully guard against this, and I think more should be done in the present paper (since model ranking is the core experiment that it runs).
Lastde++ is shown as the worst performing text detector in this paper, whereas in the Lastde paper, Lastde++ strongly outperforms Fast-DetectGPT. (Lastde++ and Fast-DetectGPT is the only pair of detection strategies covered both in the Lastde paper and the paper under review). My suspicion is that this is due to generation strategy. The RAID benchmark considers texts generated both by pure sampling from a language model probability distribution and greedy decoding. It wasn't clear to me which texts were used in this evaluation (maybe both). It seems quite plausible that greedy decoding boosts likelihood based detectors such as FastDetect-GPT more than it does detectors such as Lastde. Additionally, texts in RAID appear to have been accidentally generated using top-k sampling, since this used to be a default in huggingface (see TempTest Appendix 8.2: Local Normalization Distortion and the Detection of Machine-generated Text: Tom Kempton, Stuart Burrell, Connor J Cheverall Proceedings of The 28th International Conference on Artificial Intelligence and Statistics, PMLR 258:1972-1980, 2025. Appendix 8.2).

Let me reiterate that I think the authors have picked an interesting question, and I learned something by reading this paper, so thank you for that. But let me also stress that I think this kind of paper requires really excellent analysis of the different techniques, I don't think that patching up the criticisms above would necessarily be enough for me to recommend acceptance.

**Questions:**

Please could you detail exactly the lengths of passages of text under review.

---

> ### Author Response · Authors · 2025-11-26
> **Rebuttal by Authors (1/2)**
>
> Thank you for the valuable feedback. We truly appreciate the time and effort the reviewer invested in evaluating our work. We address the concerns below.
>
> ---
>
> ### **Justification of Our Contribution**
>
> Thank you for your feedback. Before addressing your specific concerns, we would like to clarify our contributions compared to prior studies.
>
> Regarding the transferability between MIA and machine text detection, prior literature has been **limited to specific observations**:
>
> - The mathematical equivalence found **only within a single specific pair**: DetectGPT and Neighborhood Attack [1, 2].
> - The observation that MIAs often misclassify machine-generated text as members, suggesting a potential to function as detectors (though the study **did not directly evaluate MIAs on detection** benchmarks) [2].
>
> Crucially, prior works **did not explicitly investigate the concept of transferability.** They **neither explain *why* transferability exists,** beyond noting that both rely on text probability signals, **nor do they empirically investigate *how well* methods transfer** across tasks in standard　benchmarks (i.e., whether the transferability is significant enough to be **a real issue**).
>
> We address this gap through two key contributions:
>
> 1. Theoretical contribution: We provide the **first proof that MIA and detection share the same asymptotically optimal metric**, which is the likelihood ratio between the model distribution and the true human-text distribution. This identifies **what determines performance in both tasks**: how well a method approximates the true human-text distribution, providing **practical guidance** for developing stronger methods. We also show that many popular methods can be viewed as approximations to this shared optimal metric, which **explains why transferability may arise and how it depends on the quality of this approximation**.
>
> 2. Empirical contribution: We conduct the **first extensive experiments of transferability** across 13 domains, 10 generators, and 15 representative methods, using both MIA and detection benchmarks. Our results show **strong transferability in cross-task performance**, and we further find that a machine text **detector outperforms state-of-the-art MIA methods even on the MIA** benchmark, highlighting **the need for fair evaluation** across both tasks.
>
> ---
>
> ### **Response to Weakness 1: Coverage of Detection Baselines**
>
> Thank you for the comment. We would like to clarify that our baselines **do not all focus on the same essential strategy**. As an example, both **Min-k% and Lastde (and their ++ variants)** primarily use **fluctuations in log-likelihood** (similar to FourierGPT and PHD, which you reference). These metrics also show **high transferability** but do not neatly fall into our existing categorization.
>
> ---
>
> ### **Response to Weakness 2: Justification of the Use of Rank Correlation**
>
> Thank you for raising this point. We first would like to clarify that **our proofs state that the optimal statistic is identical for both tasks**—**not that the absolute performance is correlated**. Our theoretical contribution is meant to provide intuition for **why relative rankings of methods may be preserved**, although this relationship does not directly extend to absolute values.
>
> In practice, **absolute values would be highly sensitive to benchmark-specific factors** (e.g., text length, domain, generator), which differ substantially between MIA and detection settings. Because these factors change the scale of AUROC values, absolute scores are not directly comparable and often provide limited insight into transferability. For this reason, **we believe rank correlation is a more appropriate metric in this context.**
>
> ---
>
> ### **Response to Weakness 3: Comparison Results Under Saturated Performance**
>
> Thank you for raising this concern about performance saturation leading to unreliable rankings. Specifically, we understand the concern that in a setting where all detectors perform so well (white-box), the observed differences might be attributable to random noise rather than meaningful performance distinctions.
>
> To directly address this concern, we conducted **additional statistical verification** focusing on the white-box setting. We then found that, even when numbers are so close, the **performance differences remain statistically meaningful**:
>
> - Even within this saturated gap, our **Wilcoxon signed-rank test** confirms that **the best method (Binoculars) significantly outperforms the runner-up (DetectLLM)** with *p* = 0.029. This indicates that the performance gap, while numerically small, is systematic rather than due to chance.
> - We calculated **Kendall’s Coefficient of Concordance** (W = 0.62), showing **substantial agreement** across 24 diverse conditions (8 domains × 3 generators), which demonstrates that the performance ranking is robust.
>
> We will include this analysis into our final version.

---

> ### Author Response · Authors · 2025-11-26
> **Rebuttal by Authors (2/2)**
>
> ### **Response to Weakness 4:  Clarification of the Role and Interpretation of Figure 3**
>
> Thank you for raising your valuable observation.
>
> We would like to clarify that the **primary objective of Figure 3 is to explicitly demonstrate that MIA methods achieve AUROC scores comparable to detectors** on the detection task, and vice versa. For the evaluation of the overall transferability, we **rely on the rank correlation shown in Figure 2 ($\rho > 0.6$)**, as detailed in our "Response to Weakness 2".
>
> Regarding your observation on LOSS and LogRank, we agree that their performance differs significantly across the two settings. However, we believe this behavior is line with our theoretical proof. Our unified optimality theory suggests that **transferability may correlate with how well a method approximates the optimal metric**, but it **does not imply that all methods are expected to exhibit strong transferability**. Since “pure” metrics like LOSS **do not include any component that approximate**s the true population distribution, their **lower transferability would be a theoretically expected result**, distinguishing them from sophisticated approximations like Binoculars.
>
> Furthermore, our observations suggest a potential structural factor that may enhance transferability: **self-normalization by variance**. We observe that methods incorporating this normalization (e.g., Fast-DetectGPT and Lastde++) consistently exhibit high transferability. A compelling example is the comparison between DetectGPT (=Neighborhood attack) and Fast-DetectGPT; while they share similar curvature metric, Fast-DetectGPT (with variance normalization) transfers significantly better. We hypothesize that this normalization **might help mitigate the "prior distribution difference"** between tasks (where machine text generally has lower perplexity than human text, as discussed in Section 3.3 regarding Zlib), thereby making the metric more robust to task-specific distributional shifts.
>
> ---
>
> ### **Response to Weakness 5: Consistency of Detector Rankings Across Benchmarks and Generation Settings**
>
> We agree that greedy decoding would have advantaged certain methods over others. Our evaluation used the **temperature=1, top-p=1, no repetition penalty subset of RAID**—our apologies for not including this in the paper! We suspect that **the discrepancy between the Lastde++ paper's rankings and ours may be due to their hyperparameters** (s=4, e=8n, t=15, sampling=100, etc.) **being tuned for their particular data distribution**, rather than a fundamental mistake in our evaluation setup. We agree with the reviewer that relative ranking is an important and difficult problem, but it is a problem that is neither unique to AI detection, nor within our ability to solve. If the reviewer has concerns about specific aspects of our experimental conditions, we are more than happy to discuss those.
>
> ---
>
> ### **Response to Question: Clarification of Text Passage Length**
>
> We provide the lengths of the evaluated passages are as follows. For white-box detection, the average length was 389.4 for machine and 378.1 for human. For black-box detection, the average length was 340.1 tokens for machine and 378.1 for human. For MIA, passages were filtered to >100 words and truncated to at most 200 words.
>
> ---
>
> References:
>
> [1] Shi et al. Detecting Pretraining Data from Large Language Models. ICLR 2024.
>
> [2] Naseh et al. Synthetic data can mislead evaluations: Membership inference as machine text detection. CoRR 2025.

---

### Official Review · Reviewer_EQbi · 2025-10-31

**Soundness:** 3
**Presentation:** 3
**Contribution:** 3
**Rating:** 6
**Confidence:** 4

**Summary:**

The paper systematically builds a connection between two tasks: membership inference attacks (MIAs) and machine-generated text detection (MGT detection).
It both theoretically and empirically validates that the performance of 12 methods on the two tasks is correlated across many setups.
The paper also contributes a unified evaluation suite.

**Strengths:**

1. The theoretical section presents evidence for the target similarity of the two tasks.

2. The experiments are comprehensive.

3. It is surprising to find MGT detection methods perform pretty well on MIA tasks.

**Weaknesses:**

1. The experimental setting is not perfectly aligned for the two tasks. For example, the MIA generators are all Pythia models, but Pythia is not within the MGT generators. If there is any reason for this, it should be mentioned in the paper.

2. The theoretical section mainly suggests that the absolute performance is correlated, but the presented major results are on rank correlation, which weakens the findings. One reason for this might be the unaligned experiment setting.

3. Lack of explanation of Binoculars' good performance on MIA. Is that because of the introduction of the second model (M_ref)? If so, why does DC-PDD, which also has an M_ref, not perform that well? Is that because Binoculars has an M_ref which is stronger than DC-PDD's and also the target model (Pythia)?

**Questions:**

1. Sec 2.4: It is unclear to me why "the likelihood under the true distribution is approximated by the expected likelihood of multiple perturbations." Maybe some middle steps are missed here. For example, it is assumed here assuming the model's distribution is a narrower distribution compared with the true distribution, so that perturbation, though not under the model's distribution, is still within the true distribution. (I might be wrong) More explanation is needed.

---

> ### Author Response · Authors · 2025-11-26
> **Rebuttal by Authors**
>
> Thank you for the valuable feedback. We truly appreciate the time and effort the reviewer invested in evaluating our work. We address the concerns below.
>
> ---
>
> ### **Response to Weakness 1: Alignment of Experimental Settings Across Tasks**
>
> Thank you for pointing out this fair point. Our **primary motivation** for this setup was to **assess practical utility by adhering to the most widely used benchmarks in each domain** (MIMIR benchmark for MIAs, RAID benchmark for detection). While this introduces a model discrepancy, we would expect our results to **establish a lower bound on the expected transferability when the model is the same** across both tasks.
>
> ---
>
> ### **Response to Weakness 2: Justification of the Use of Rank Correlation**
>
> Thank you for raising this point. We first would like to clarify that **our proofs state that the optimal statistic is identical for both tasks—not that the absolute performance is correlated**. Our theoretical contribution is meant to provide intuition for **why relative rankings of methods may be preserved**, although this relationship does not directly extend to absolute values.
>
> As for the distinction between relative and absolute rankings, since **absolute values are highly sensitive to benchmark-specific factors** (e.g., text length, domain, generator) as you noted, **we believe rank correlation is a more appropriate metric** in this context. Crucially, the observation that these rankings are **highly preserved (ρ=0.66) even under such unaligned settings** provides **strong evidence for the robustness of our theoretical claims**.
>
> ---
>
> ### **Response to Weakness 3: Explaining Strong MIA Performance of Binoculars**
>
> Regarding your question about whether Binoculars performs well on MIA **because its reference model is stronger than the one used in DC-PDD, your interpretation is correct**. In section A.2 we explain that $M_{ref}$ in DC-PDD represents the Laplace-smoothed unigram token frequency distribution under some reference corpus $D'$. Viewing this through the lens of our optimal statistic, we would argue that **the method performs poorly because $M_{ref}$ is a relatively poor approximation of the true distribution**, at least compared to another language model.
>
> ---
>
> ### **Response to Question: Clarification of the Perturbation-Based Approximation**
>
> Thank you for highlighting this. We view the expected likelihood of multiple perturbations as a **approximate marginalization over semantically equivalent paraphrases**. Assuming that **the average delta of likelihood between these paraphrases is larger in $M$ than it is in $Q$** (this is equivalent to the negative curvature assumption from DetectGPT), applying this marginalization to a particular datapoint would result in a **better approximation of the true distribution $Q$**. We will add this explanation to the paper.

---

> > ### Comment · Reviewer_EQbi · 2025-11-26
> > **Thank you**
> >
> > Thank you for the response! At this time, I maintain my assessment (rating=6) unchanged.

---

> > > ### Author Response · Authors · 2025-11-28
> > >
> > > Thank you for letting us know, and we appreciate your time and assessment!
> > > If you have any additional questions or would like further clarification on any aspect of the work, we would be more than happy to discuss them.

---

### Official Review · Reviewer_U6kd · 2025-11-01

**Soundness:** 2
**Presentation:** 3
**Contribution:** 2
**Rating:** 2
**Confidence:** 4

**Summary:**

In this paper, the authors argue that membership inference attacks (MIAs) on large language models are equivalent, at least in the limit, to LLM text detection, and that the In order to support their argument they focus on two statistical tests and decision functions, one used by MIA method, the Neighbourhood Attack, and one - by a common LLM text detector, Binoculars, and prove their equivalence. Additionally, they prove that in the limit, the functions used by these decision functions are asymptotically optimal, at least with regard to a fixed false positive rate. The authors then validate their hypothesis by experimentally comparing the performance on a selection of MIAs and LLM detectors on both MIA and LLM tasks, and claim additional contribution by publishing a comprehensive mixed benchmark.

**Strengths:**

Equivalence and optimality proofs are always valuable in the context of algorithm development, given their ability to provide definitive and incontestable insights into the performance of different method classes. The authors' focus on Type I error, most concerning in the context of LLM detectors, in their proofs is a strong point, making this paper more relevant to the LLM detectors community.

**Weaknesses:**

I am not convinced of the interest or contribution of this paper, at least not to a level that warrants acceptance to a conference such as ICLR.

Specifically:
- The author's theory hinges on the assumption that texts from its training dataset are generated by a model with a lower perplexity. While generally assumed, more recent results suggest that this might not be the case, with recent findings [1] indicating that even low-perplexity LLM-generated sequences do not map directly to the training data.
- For LLM detectors, no evasion attacks are considered, despite being a critical problem in that field, as highlighted by the RAID benchmark paper authors cite themselves. In that setting, the models are known to be generating texts that are not present in their training dataset, and the problem of detection cannot be equivalent to the MIA problem, at least to the best of my understanding.
- For MIA attacks, the authors are using a dataset known to induce post-hoc datasets with a positive bias, which has been criticized in the past as a poor proxy for the detection tasks, namely by [2].
- I am not convinced of the value of the proofs. As mentioned by the authors, MIA and LLM text detection have been understood to be related. However, I am not convinced that a proof of equivalence for a specific MIA and a specific LLM detector with convenient detection functions adds value to the field.
- Additionally, validation of the claims regarding the benchmark publication was impossible to validate, given that the linked repository returned file-not-found errors for all Python files in the directory structure except the top-level one.

[1] Low-Perplexity LLM-Generated Sequences and Where To Find Them (https://aclanthology.org/2025.acl-srw.51/) Wuhrmann et al., ACL 2025
[2] Meeus, M., Shilov, I., Jain, S., Faysse, M., Rei, M., & Montjoye, Y.D. (2024). SoK: Membership Inference Attacks on LLMs are Rushing Nowhere (and How to Fix It). 2025 IEEE Conference on Secure and Trustworthy Machine Learning (SaTML), 385-401.

**Questions:**

- What is the difference between black-box and white-box models for detectors? To the best of my understanding, none of them rely on text perplexities during the generation by design and treat both settings in the same fashion.

- Why did you choose to use AUC as a metric, despite citing the landmark (Carlini et al. 2022) paper specifically arguing in the abstract for the use of TPR at fixed low FPR?

- Could you please provide common information regarding experimental set-up (hardware, OS & drivers version, requirements.txt, and versions of unfrozen libraries), CO2 emissions estimation, GenML tools usage, ...

---

> ### Author Response · Authors · 2025-11-26
> **Rebuttal by Authors (1/2)**
>
> Thank you for the valuable feedback. We truly appreciate the time and effort the reviewer invested in evaluating our work. We address the concerns below.
>
> ---
>
> ### **Justification of Our Proof Focus**
>
> Regarding the statement in **the summary about us “proving the equivalence and asymptotic optimality of Neighborhood Attack and Binoculars,”** allow us to clear up a **misconception** here.
>
> We do not prove that the Neighborhood Attack and Binoculars are equivalent nor do we prove they are asymptotically optimal. **Rather, we prove that the optimal decision function —** $\frac{L(x;M)}{L(x;Q)}$ **— is the same for both tasks**, and that many methods from both tasks can be viewed as approximating this optimal decision function, thus explaining the oddly high degree of transferability between the two tasks.
>
> ---
>
> ### **Response to Weakness 1: Assumption on Lower Perplexity for Training Data**
>
> We thank the reviewer for pointing us to the recent findings about “low-perplexity LLM-generated sequences not mapping directly to training data”. However, **we are not quite sure what the reviewer means when they claim that our theory “hinges on the assumption that texts from its training dataset are generated by a model with a lower perplexity”**. **We do not claim this in our work**. Please allow us to clarify our interpretation of the concern:
>
> We interpret the reviewer’s concern to be that, when machine-generated text $x$ is high likelihood (e.g. $L(x;M) > L(x;Q)$) and not in the training set, our proposed statistic will correctly classify $x$ as machine-generated text but incorrectly classify it as being in the training data, thus serving as a counterexample for our statistic being optimal for both tasks. (Please let us know if this is the correct interpretation of the comment.)
>
> Under the initial conditions of our proof, **the phenomenon detailed by [1] cannot happen**. Our proof requires $M$ to be trained under standard asymptotic conditions (MLE objective, sufficient model capacity, arbitrarily many draws from the training distribution). Thus the maximum likelihood according to $M$ of any particular sample outside of the training data can be arbitrarily minimized during training. This means that all text generated by $M$ with sufficiently low-perplexity will always map directly to the training data and the phenomenon from [1] is not possible.
>
> Of course, in non asymptotic settings where we have limited draws from our training set, it may be the case that machine-generated text is more likely than certain training documents. In these cases, the proposed statistic will misclassify the example in at least one of the two tasks. However, **our proof does not apply in the non-asymptotic case and therefore this does not contradict our core claim**.
>
> ---
>
> ### **Response to Weakness 2: Evasion Attacks and Task Equivalence**
>
> Thank you for the insightful comment. Our objective is to examine how well methods developed for one task transfer to the other. To this end, evaluating both tasks under **a common non-adversarial setup is the most direct and appropriate way** to reveal their methodological connections. We are **not claiming that detection and MIA are equivalent problems** (this is why our title is “*Machine Text Detectors are MIAs*” rather than “*Machine Text Detection is Membership inference.*”). MIAs and detectors require fundamentally different robustness profiles in order to be useful in practical settings. Our work simply claims that, **although the two tasks are different problems** with different goals, they **fundamentally rely on the same underlying signal** and **importantly share the same asymptotically optimal metric**. Our extensive experiments support this empirical connection by demonstrating strong transferability of methods across tasks.
>
> That said, **we agree that the nature of the adversarial attacks present in both tasks is a crucial distinction between detection and MIAs** and is relevant for practical deployment. We will add a discussion of this point in the paper.
>
> ---
>
> ### **Response to Weakness 3: Bias in the MIA benchmark**
>
> Thank you for pointing out the potential bias in the MIA benchmark we used. As noted in Section 3.1, we **intentionally selected the MIMIR benchmark with 13-gram deduplication**, which **the cited paper [2] identifies as** **the reliable configuration that ensures evaluation validity**, unlike the overly aggressive 7-gram deduplication.

---

> ### Author Response · Authors · 2025-11-26
> **Rebuttal by Authors (2/2)**
>
> ### **Response to Weakness 4: Value of Our Theoretical Proofs**
>
> We would like to clarify that **our proofs are not derived from any specific MIA or detector, but follow directly from the task formulation**. When both tasks are framed as hypothesis testing, the Neyman–Pearson lemma shows that the likelihood ratio test between a target language model and the true human-text distribution is **the uniformly most powerful static for both tasks**. In Section 2.4, we also show that **many popular methods can be viewed as approximations to this shared optimal metric**, which explains why their transferability depends on how well they approximate it.
>
> Our proofs further **identify what determines performance in both tasks**: how well a method approximates the true human-text distribution. This insight provides **practical guidance for developing stronger methods**. Although **transferability has been discussed only briefly** (e.g., noting a single identical pair or briefly observing that MIAs misclassify machine-generated non-members, but they didn't evaluate MIA on detection benchmark), we are **the first to provide a theoretical explanation consistent with extensive experimental results**, highlighting **the importance of fair evaluation across tasks**.
>
> The motivating pair (Neighborhood Attack vs. DetectGPT) is included **only as an intuitive introduction**, helping readers see why transferability is a meaningful question.
>
> ---
>
> ### **Response to Weakness 5: Repository Access Issue**
>
> The repository hosted on Anonymous GitHub (https://anonymous.4open.science/) is supposed to be open access, and **we have confirmed from our side** (across multiple devices and networks) that **it is accessible**. It is possible that the issue was temporary or environment-specific, so we kindly ask that you try accessing it again.
>
> ---
>
> ### **Response to Question 1: Difference Between Black-box and White-box Detectors**
>
> As stated in Section 3.1, the distinction depends on whether the target model **allows access to its token probability distribution**. When accessible, we consider the setting to be white-box. When they are not accessible like ChatGPT, the setting is black-box. In the **black-box setting, we employ surrogate models to compute logits**. We **observe transferability in both settings**.
>
> ---
>
> ### **Response to Question 2: Justification of the Use of AUROC**
>
> We believe that evaluating **transferability based on performance at a single threshold is not reliable**, because it depends heavily on the particular choice of threshold and may not reflect the overall behavior of a method across the two tasks. For this reason, **we assess transferability using AUROC**, which **captures the detection performance across all possible thresholds**.
>
> ---
>
> ### **Response to Question 3: Hardware, Software, and Environmental Details**
>
> We provided the requested information below:
>
> **GenML tools usage:**
> We used GPT5 for help drafting sections of the paper as well as during ideation. We also used GitHub Copilot during code writing, especially for minor code completion.
>
> **Hardware and software setup:**
> GPU: 4 × NVIDIA RTX A6000 (48GB), CPU: Intel Xeon Silver 4314, RAM: 300GB, OS: openSUSE Leap 15.6, NVIDIA Driver: 560.28.03, CUDA version: 12.6
>
> **CO₂ emissions estimate:**
> Our experiments used approximately 350 GPU-hours on RTX A6000 GPUs. Based on the ML CO2 Impact Calculator (assuming a global average PUE of 1.58), this corresponds to roughly 66kg CO2 equivalent emissions.
>
> **requirements.txt:**
>
> ```python
> datasets==3.2.0
> matplotlib==3.9.3
> numpy==1.25.2
> pandas==2.0.3
> raid-bench==0.0.9
> requests==2.32.3
> scikit-learn==1.3.2
> tiktoken==0.8.0
> torch==2.5.1
> tqdm==4.67.1
> transformers==4.47.0
> accelerate==1.6.0
> zstandard==0.25.0
> ```
> ---
>
> References:
>
> [1] Wuhrmann et al. Low-Perplexity LLM-Generated Sequences and Where To Find Them. ACL 2025 SRW.
>
> [2] Meeus et al. SoK: Membership Inference Attacks on LLMs are Rushing Nowhere (and How to Fix It). SaTML 2025.

---

### Meta-Review · Area_Chair_tp8G · 2026-01-06

**Summary:**

This paper studies the connections between the task of MIA and AI-text detection. In other words, they theoretically and empirically investigate the transferability. Basically, it lacks of novelty. It is well-known these two tasks are highly related. proving equivalence for a convenient pair may not add much. Proof value is limited / not novel. It is raised by: U6kd, also EPu5

Two reviewers give strong negative feedbacks. From the rebuttal, I do not think they will change their mind.

Suggestions as also raised by EPu5, the detection baselines incomplete / too concentrated on likelihood-style methods. It is suggested to incude more detection approaches, such as:
DNA-GPT: Divergent N-Gram Analysis for Training-Free Detection of GPT-Generated Text, ICLR'24.
DALD: Improving Logits-based Detector without Logits from Black-box LLMs. NeurIPS'24.
Radar: Robust ai-text detection via adversarial learning. NeurIPS'23.
Detective: Detecting ai-generated text via multi-level contrastive learning. NeurIPS'24.
Human Texts Are Outliers: Detecting LLM-generated Texts via Out-of-distribution Detection. NeurIPS'25.

**Reviewer Concerns:**

1. It is well-known these two tasks are highly related. Proving equivalence for a convenient pair may not add much. Proof value is limited / not novel. It is raised by: U6kd, also EPu5

2. Metric choice: AUROC vs TPR @ fixed low FPR. It is raised by: U6kd
Authors’ response: They justify AUROC because transferability based on a single threshold is unstable; AUROC summarizes across thresholds.
For detection, i think TPR @ fixed low FPR is more important.

**Reviewer Scores:**

Half positive, half strong negative. no evidence the two negative ones are willing to update scores.

---

### Decision · Program_Chairs · 2026-01-26

Reject